



# Oxidative capacity and radical chemistry in the polluted atmosphere of Hong Kong and Pearl River Delta region: analysis of a severe photochemical smog episode

Likun Xue[1*], Rongrong Gu[1], Tao Wang[2,1], Xinfeng Wang[1], Sandra Saunders[3], Donald Blake[4], Peter K. K. Louie[5], Connie W. Y. Luk[5], Isobel Simpson[4], Zheng Xu[1], Zhe Wang[2], Yuan Gao[2], Shuncheng Lee[2], Abdelwahid Mellouki[1], and Wenxing Wang[1]

[1] Environment Research Institute, Shandong University, Ji'nan, Shandong, China

[2] Department of Civil and Environmental Engineering, Hong Kong Polytechnic University, Hong Kong, China

[3] School of Chemistry and Biochemistry, University of Western Australia, WA, Australia

[4] Department of Chemistry, University of California at Irvine, Irvine, CA, USA

[5] Environmental Protection Department, the Government of Hong Kong Special Administrative Region, Hong Kong, China

*To whom correspondence should be addressed

E-mail: xuelikun@sdu.edu.cn, Tel: +86-531-8836 1185

## Abstract

We analyze a multi-day photochemical smog episode to understand the oxidative capacity and radical chemistry of the polluted atmosphere in Hong Kong and the Pearl River Delta (PRD) region. A photochemical box model based on the Master Chemical Mechanism (MCM v3.2) is constrained by an intensive set of field observations to elucidate the budgets of $RO_X$ ($RO_X=OH+HO_2+RO_2$) and $NO_3$ radicals. Highly abundant radical precursors (*i.e.,* $O_3$, HONO and carbonyls), nitrogen oxides ($NO_X$) and volatile organic compounds (VOCs) facilitate strong production and efficient recycling of $RO_X$ radicals. The OH reactivity is dominated by oxygenated VOCs (OVOCs), followed by aromatics, alkenes and alkanes. Photolysis of OVOCs (except for formaldehyde) is the dominant primary source of $RO_X$ with an average daytime contribution of 47%. HONO photolysis is the largest contributor to OH and the second most significant source (19%) of $RO_X$. Other considerable $RO_X$ sources include $O_3$ photolysis (11%), formaldehyde photolysis (10%), and ozonolysis reactions of unsaturated VOCs (6.2%). In one case when solar irradiation was attenuated by the high aerosol loadings, $NO_3$ became an important oxidant and the $NO_3$-initiated VOC oxidation presented another significant $RO_X$ source (6.2%) even during daytime. Sensitivity studies show that controlling aromatics is the most efficient way to reduce the atmospheric oxidative capacity and mitigate photochemical pollution in Hong Kong. This study suggests the possible impacts of daytime $NO_3$ chemistry in polluted atmospheres under conditions with the co-existence of abundant $O_3$, $NO_2$, VOCs and aerosols, and also provides new insights into the


radical chemistry that essentially drives the formation of photochemical smog in Hong Kong and the PRD region.

## 1. Introduction

The hydroxyl radical (OH) and hydro/organic peroxy radicals ($HO_2$ and $RO_2$), collectively known as ROx, play a central role in atmospheric chemistry and air pollution (Stone et al., 2012). They dominate the oxidative capacity of atmosphere, and hence govern the removal of primary contaminants and formation of secondary pollutants such as ozone ($O_3$) and secondary organic aerosol (Hofzumahaus et al., 2009). In the troposphere, they arise from photolysis of closed-shell molecules such as $O_3$, nitrous acid (HONO), formaldehyde (HCHO) and other carbonyls, as well as ozonolysis reactions of unsaturated volatile organic compounds (VOCs) (Dusanter et al., 2009; Lu et al., 2012; Volkamer et al., 2010). In the presence of nitrogen oxides ($NO_X$) and VOCs, the $RO_X$ radicals can undergo efficient recycling (*e.g.,* OH→$RO_2$→ RO→$HO_2$→OH) and produce $O_3$ and oxygenated VOCs (OVOCs) (Sheehy et al., 2010). The radical recycling is terminated by their cross reactions with $NO_X$ (under high $NO_X$ conditions) and $RO_X$ themselves (under low $NO_X$ conditions), which results in the formation of nitric acid, organic nitrates and peroxides (Liu et al., 2012; Wood et al., 2009). Given the essential significance and complex processes involved, radical chemistry presents one of the core areas in the atmospheric chemistry research.

Understanding the sources and chemistry of $RO_X$ has long been a focus of air quality studies over the past decades. It has been shown that although air pollution problems are visually quite similar, the radical chemistry, and in particular the relative importance of primary radical sources, is inhomogeneous in different metropolitan areas. For example, the dominant radical sources are $O_3$ photolysis in the South Coast Air Basin in California (2010 scenario) and Nashville, US (Martinez et al., 2003; Griffin et al., 2004; Volkamer et al., 2010); HONO photolysis in New York City, US (Ren et al., 2003), Paris, France (Michoud et al., 2012) and Santiago, Chile (Elshorbany et al., 2009); HCHO photolysis in Milan, Italy (Alicke et al., 2002); and OVOC photolysis in Mexico City, Mexico (Volkamer et al., 2010), Beijing, China (Liu et al., 2012), Birmingham (summer scenario; Emmerson et al., 2005) and London in England (Emmerson et al., 2007). Therefore, identification of the principal radical sources is a fundamental step towards understanding the formation of air pollution and formulating science-based control strategies.

The nitrate radical ($NO_3$) is another important oxidant in the polluted atmosphere (Geyer et al., 2001). The $NO_3$-initiated degradation of VOCs presents an important source of $RO_2$, gaseous organic nitrates and nitrogen-containing aerosols (Rollins et al., 2012; Saunders et al., 2003). $NO_3$ has been recognized as a major player in nocturnal chemistry, but is usually neglected for the daytime chemistry given its fast photolysis in sunlight (Volkamer et al., 2010). Under certain conditions, *e.g.,* with abundant $O_3$ and $NO_2$ (hence strong $NO_3$ production) and weak solar radiation (thus weak photolysis),





however, $NO_3$ may also play a role in the daytime chemistry. Indeed, Geyer et al. (2003) observed by differential optical absorption spectroscopy (DOAS) ~5 pptv of $NO_3$ three hours before sunset in Houston, and indicated considerable contribution (10%) of $NO_3$ chemistry to the daytime $O_X$ loss. More studies are required to confirm the possible operation of $NO_3$ chemistry during daytime and to evaluate its impacts on the atmospheric oxidative capacity (AOC) and formation of $O_3$ and secondary aerosols.

Hong Kong and the adjacent Pearl River Delta (PRD) is the most industrialized region of southern China, and is suffering from serious photochemical air pollution (e.g., Ling et al., 2014; Zheng et al., 2010). A number of studies have been conducted in the last decade, most of which focused on either $O_3$-precursor relationships (Zhang J. et al., 2007; Zhang Y. et al., 2008) or local vs. regional contributions (Wang et al., 2009; Li et al., 2012; Xue et al., 2014b), but few has attempted to understand the atmospheric oxidizing capacity and radical chemistry (Lu et al., 2014). Recent studies have observed the highest ever-reported concentrations of OH and $HO_2$ at a rural site in the northern PRD, which cannot be reproduced by the classic knowledge of atmospheric chemistry (Hofzumahaus et al., 2009). This indicates the strong oxidative capacity of the atmosphere in this region as well as a deficiency in understanding the chemistry underlying the pollution.

As part of the Hong Kong Supersite programme aimed at elucidating the causes of regional smog and haze pollution, an intensive field campaign was conducted at a regional receptor site in summer 2011. A comprehensive set of measurements was taken, which facilitated the construction of a detailed observation-constrained box model to study the atmospheric photochemistry. In the present work, we analyze a severe 7-day photochemical episode (25–31 August 2011) to gain an understanding of atmospheric oxidative capacity and radical chemistry. We first provide an observational overview of the episode, and then evaluate the chemical budgets of both $RO_X$ and $NO_3$ radicals. A series of sensitivity studies were performed to examine the impacts of the controllable photochemical precursors, *i.e.,* $NO_X$ and VOCs, on the atmospheric oxidative capacity and radical sources. This study provides some new insights regarding: (1) the potential impact of $NO_3$ on the daytime photochemistry in polluted atmospheres and (2) the primary radical sources of $RO_X$ in Hong Kong and the PRD region.

## 2. Methods

### 2.1. Experimental

The measurements were conducted at the Tung Chung air quality monitoring station (TC; 113.93 ° E, 22.30 ° N). It is located about 3 *km* south of the Hong Kong International airport, and is in a residential area of a new town in western Hong Kong (see *Fig. S1*). This station is characterized as a polluted receptor site as it receives urban plumes from Hong Kong under easterly winds and regional air masses from the PRD region when northerly winds prevail, and is the location where the maximum $O_3$ levels are usually recorded in Hong Kong (Xue et al., 2014b). Details of this station and analyses of



HONO and aerosol data have been described in our previous publications (Xu et al., 2015; Xue et al., 2014b; Zhou et al., 2014).

A one-month campaign was carried out from 6 August to 7 September 2011, which covered two distinct types of meteorological conditions and air quality (see *Figure S2*). For the majority of the campaign, southerly winds dominated bringing clean marine air masses and good air quality (typical summer conditions as a result of the Asian monsoon). In contrast from 25-31 August, a heavy multi-day photochemical smog event hit Hong Kong with northerly winds prevailing during the daytime and elevated concentrations of various air pollutants were observed. In the present study, this episode was subject to a detailed modeling analysis to understand the atmospheric oxidative capacity and $RO_X$ chemistry, made possible with the most comprehensive suite of measurements taken for the first time in Hong Kong.

A full suite of trace gases and meteorological parameters were simultaneously measured during this episode (as summarized in *Table S1*). Here a brief description is given of the measurements used in the present study. Major air quality target pollutants were routinely monitored with commercial analyzers: $O_3$ with a UV photometric analyzer (*TEI model 49i*); CO with a non-dispersive infrared equipment (*API model 300EU*); NO and $NO_2$ with a chemiluminescence analyzer (*TEI model 42i*) equipped with a selective blue light converter (Xu et al., 2013). $NO_y$ was measured by another chemiluminescence instrument (*TEI model 42cy*) with an external MoO catalytic converter (Xue et al., 2011). HONO was measured in real-time by a long path absorption photometer (*QUMA model LOPAP-03*) (Xu et al., 2015). Nitryl chloride ($ClNO_2$) was detected using a custom-built chemical ionization mass spectrometer (CIMS; *THS Instruments Inc., Atlanta*) (Tham et al., 2014). Peroxyacetyl nitrate (PAN) was measured by the same CIMS instrument with a heated inlet, and the potential interference caused by high NO was corrected based on laboratory tests (Slusher et al., 2004; Wang et al., 2014). Hydrogen peroxide ($H_2O_2$) and organic peroxides were measured by an enzyme-catalyzed fluorescence instrument (*Aerolaser AL-2021*) (Guo et al., 2014). Particle number and size distributions in the range of 5 nm – 10 μm, which were used to calculate the aerosol surface density, were measured with a wide range particle spectrometer (*WPS; MSP model 1000XP*) (Gao et al., 2009).

$C_2-C_{10}$ non-methane hydrocarbons were measured at a time interval of 30 minutes by a commercial analyzer that combines gas chromatography (GC) with photoionization detection (PID) and flame-ionization detection (FID) (*Syntech Spectras, model GC955 Series 600/800 POCP*). In addition, 24-hour whole air canister samples were collected on selected days (e.g., 25 and 29 August) for the detection of $C_1-C_{10}$ hydrocarbons by using GC with FID, electron capture detection (ECD) and mass spectrometry detection (MSD), with the analyses carried out at the laboratory of the University of California at Irvine (Simpson et al., 2010; Xue et al., 2013). As evaluated in our previous study, both sets of hydrocarbon measurements agree very well apart from the alkenes. Here the real-time data



tended to systematically overestimate the canister measurements (Xue et al., 2014b). Hence, in the present study, the high resolution real-time data were corrected according to the canister data. $C_1$-$C_8$ carbonyls were measured by collecting air samples on DNPH-coated sorbent cartridges followed by high pressure liquid chromatography analysis (Xue et al., 2014c). For the carbonyls, a 24-hour

integrated sample was collected on 25 August, and eight 3-hour samples were taken throughout the day on 31 August. The measured hydrocarbon and carbonyl species are listed in *Table 1*.

Meteorological parameters were monitored by a series of commercial sensors, including a probe for ambient temperature and relative humidity (*Young RH/T probe*) and an ultrasonic sensor for wind speed and direction (*Gill WindSonic*). Photolysis frequency of $NO_2$ ($J_{NO2}$) was measured with a filter

radiometer (*Meteorologie Consult gmbh*). All of the above techniques have been validated and applied in many previous studies, with detailed descriptions of the measurement principles, quality assurance and control procedures provided elsewhere (Guo et al., 2014; Xu et al., 2015; Xue et al., 2011, 2014a, 2014b and 2014c). See also *Table S1* for a summary of the measurement techniques/instruments and time resolutions.

**2.2.  The OBM-AOCP model**

The zero-dimensional chemical box model OBM-AOCP (Observation-Based Model for investigating the Atmospheric Oxidative Capacity and Photochemistry) has been utilized in many previous studies to evaluate $O_3$ production (Xue et al., 2013, 2014a, 2014b), PAN formation (Xue et al., 2014c), and oxidative capacity (Xue et al., 2015). Briefly, the model is built on the Master Chemical

Mechanism (MCM; v3.2), a nearly explicit gas phase mechanism describing the degradation of 143 primary VOCs (Jenkin et al., 2003; Saunders et al., 2003), and is updated to include both a heterogeneous chemistry scheme (including heterogeneous processes of $NO_2$, $NO_3$, $N_2O_5$, $HO_2$ and $ClONO_2$; Xue et al., 2014a) and a chlorine chemistry module that describes the reactions of Cl radical with various VOC compounds (Xue et al., 2015; note that the basic MCM only considers the reactions

of Cl radical with alkanes). In addition to the chemistry, dry deposition and dilution mixing within the boundary layer are also included in the model (Xue et al., 2014a). The mixing layer height affecting the deposition rate and dilution mixing was assumed to vary from 300 m at night to 1500 m in the afternoon. Sensitivity model runs with different maximum mixing heights (1000 and 2000 m) indicated that its impacts on the modeling results were negligible.

The model is capable of simulating the concentrations of highly reactive species (*e.g.,* radicals) and quantitatively evaluating several key aspects of atmospheric photochemistry such as oxidant formation (*e.g.,* $O_3$ and PAN), VOC oxidation and radical budgets. The rates of over 15600 reactions are individually and instantaneously computed within the model and grouped into a relatively small number of major routes. The calculation of ozone and PAN production rates have been described elsewhere

(Xue et al., 2014a; 2014c). Here the emphasis is placed on the computation of AOC and $RO_X$ budget.



AOC is calculated as the sum of oxidation rates of CO and VOCs by the principal oxidants, namely OH, $O_3$, $NO_3$ and Cl (Xue et al., 2015). The partitioning of the AOC among individual oxidants or VOC groups can be also assessed. The chemical budgets of OH, $HO_2$ and $RO_2$ are quantified by grouping a huge number of relevant reactions into dozens of major production, cycling and loss routes. The

principal radical sources in the polluted atmosphere generally include photolysis of $O_3$, HONO, $H_2O_2$ and OVOCs as well as reactions of $O_3$+VOCs, $NO_3$+VOCs and Cl+VOCs. The radical sinks mainly include the $RO_X$-$NO_X$ and $RO_X$-$RO_X$ cross reactions. Besides, a number of other minor reaction pathways were also computed to facilitate a thorough investigation of the $RO_X$ chemistry (see *Figures 6 and 7*).

The measurement data of $O_3$, NO, $NO_2$, HONO, $ClNO_2$, $H_2O_2$, PAN, CO, $C_1$-$C_{10}$ HCs, $C_1$-$C_8$ carbonyls, aerosol surface area and radius, temperature, RH and $J_{NO2}$ were averaged or interpolated to a time resolution of 10 minutes for the model constraints. For carbonyls, the diurnal profiles measured on 31 August 2011, throughout which eight 3-hour samples were collected, were adopted and scaled to the 24-hour average data observed on 25 August (see *Figure S3* for the measured profiles of selected

carbonyls). Photolysis frequencies were calculated as a function of solar zenith angle within the model (Saunders et al., 2003) and further scaled with the measured $J_{NO2}$ values. The model starts from 00:00 local time and runs for a 24-hour period. Prior to formal simulation, the model was run for five days with constraints of the campaign-average data to produce reasonable estimates of initial conditions for the unconstrained compounds (*e.g.,* radicals). The final outputs were extracted and subject to further

analyses.

## 3. Results and Discussion

### 3.1. Observational overview

During 25-31 August 2011, Hong Kong and the adjacent PRD region was hit by a prolonged smog episode, with concentrations of various air pollutants exceeding ambient air quality standards as

recorded at many air quality monitoring stations over the region. The measured concentrations of major pollutants and meteorological parameters at TC are depicted in *Figure 1*. Peak $O_3$ mixing ratios of over 150 ppbv were observed almost every day within the 1-week period, except for 29 August when the peak was 135 ppbv. As another indicator of photochemical smog, the concentrations of PAN were also very high with the peak values exceeding 4 ppbv every day (except for 3.7 ppbv on 29 August). The

maximum hourly values of $O_3$ and PAN were recorded at 162 and 6.95 ppbv, respectively. Meanwhile, extremely high levels of $NO_X$, $NO_y$ (peaks of ~150 ppbv), CO (peak of ~1000 ppbv) and $SO_2$ (peak of ~20 ppbv) were also determined. Overall, inspection of the data reveals the markedly poor air quality and intense photochemical oxidant production over the region during the episode.

*Table 1* lists the 24-h average concentrations of hydrocarbons and carbonyls measured on 25 and 31



August 2011. It is clearly seen that the VOC levels, in particular for reactive aromatics and aldehydes, were also very high during the episode. On 25 August, for instance, the 24-h average values of toluene, summed xylenes, formaldehyde and acetaldehyde were as high as 9.47, 3.87, 9.89 and 4.25 ppbv, respectively. HONO and $ClNO_2$, two precursors of OH and Cl radicals, were also measured. Elevated

HONO (up to 2–3 ppbv) and moderate $ClNO_2$ (up to 0.5–1 ppbv) were usually found at night, and what is more interesting is that the daytime HONO levels were also significant (over 1 ppbv in general; see *Fig. 1*). Such daytime HONO levels cannot be explained by the known gas-phase source and indicates the existence of other unknown source(s) (Xu et al., 2015), yet exploring the unknown HONO sources is beyond the scope of the present study. High abundances of $O_3$, HONO and carbonyls would definitely

lead to strong production of $RO_X$ radicals, and the abundant VOCs would facilitate efficient radical recycling. Therefore, strong atmospheric oxidative capacity and intensive *in-situ* photochemistry can be expected from the above analyses.

The dynamic cause of this episode was a distant tropical cyclone that introduced warm stagnant weather and facilitated accumulation of air pollutants in Hong Kong and the PRD region (Ding et al.,

2004). The weather condition featured high temperatures (30 ℃–35 ℃) and relatively low RH (40%– 80%; see *Fig. 1*). During the daytime, the prevailing surface winds were consistently from the northwest with relatively low wind speeds (~2 m/s), suggesting the transport of processed air masses from the upwind PRD region to the site. This was further confirmed by the 48-h backward trajectories calculated by the HYSPLIT model (Draxler and Rolph, 2016), which indicated that for most days the air masses

had spent a large portion of time over the PRD region prior to arriving at TC (*Fig. 2*).

There was an exception on 25 August when the air flow was switching from southerly maritime air to northerly PRD regional air masses (see *Fig. S2*). This case is believed to be more influenced by the local air in Hong Kong, because (1) northerly winds during the daytime were somewhat weak compared to the other cases (see *Fig. 1*); (2) the backward trajectories also indicated less impact from the PRD

region (*Fig. 2*); and (3) the $CO/NO_y$ ratio on that day was significantly lower than those on the following days (*Fig. S4*), which is consistent with the previous finding that the PRD air masses have higher $CO/NO_y$ ratios than those from Hong Kong (Wang et al., 2003). The evolution of the $CO/NO_y$ ratio clearly indicates the transition from local (25 August) to regional air masses (27–31 August) throughout the 1-week episode (*Fig. S4*). In the following discussion, detailed modeling analyses are

conducted for the 25 and 31 August cases, which are representative of local Hong Kong and regional PRD pollution, respectively.

### 3.2.  Atmospheric oxidative capacity

The strong oxidative capacity of the atmosphere during the pollution episodes was confirmed by quantifying the loss rates of CO and VOCs via reactions with OH, $O_3$, $NO_3$ and Cl, as shown in *Fig. 3*.

The calculated AOC was up to $2.04 \times 10^8$ and $1.27 \times 10^8$ molecules $cm^{-3}$ $s^{-1}$, with daytime averages (6:00–



18:00 LT) of $7.26 \times 10^7$ and $6.30 \times 10^7$ molecules $cm^{-3} s^{-1}$, on 25 and 31 August, respectively. As such, the total number of CO and VOC molecules depleted throughout the daytime was $3.14 \times 10^{12}$ and $2.72 \times 10^{12}$ per $cm^{-3}$ of air for both cases. OH was, as expected, the predominant oxidant accounting for 89% and 93% of the AOC on 25 and 31 August, respectively. $NO_3$ was the second most important oxidant, particularly

on 25 August contributing on average 7% of the AOC with a maximum contribution of 43% at 15:00 LT under a weak solar radiation condition (a detailed evaluation of the role of $NO_3$ in the daytime chemistry is presented in Section 3.4). In comparison, $O_3$ and Cl had minor contributions due to the relatively lower abundances of alkenes and Cl radicals.

We further assessed the loss rates of major VOC groups due to OH oxidation, which elucidated the

partitioning of the OH reactivity among oxidation of different VOCs. The results are presented in *Fig. 4*. OVOCs clearly dominate the OH reactivity with daytime average contributions of 60% and 75% and with maximums in the afternoon of over 80% for both cases. Aromatics are the second largest contributor comprising on average 22% and 10% of the daytime OH reactivity. For the Hong Kong local case on 25 August, especially, aromatics made up the majority (*i.e.,* 40%–60%) of the OH

reactivity in the early morning period when there were much fresher air masses. In comparison, alkenes and alkanes only accounted for a small fraction (8%–10%) of the OH reactivity at TC. These results are in fair agreement with the previous study of Lou et al. (2010), which indicated the dominance of secondary OVOCs in the observed OH reactivity in the PRD region.

*Figure 5* shows the model-predicted daytime concentration profiles of OH, $HO_2$ and $RO_2$ at TC

during the two episodes. The maximum concentrations of OH, $HO_2$ and $RO_2$ were $6.4 \times 10^6$, $7.7 \times 10^8$ and $9.2 \times 10^8$ molecules $cm^{-3}$ (equivalent as 0.27, 32 and 39 pptv) on 25 August, and were $7.0 \times 10^6$, $5.3 \times 10^8$ and $4.8 \times 10^8$ molecules $cm^{-3}$ (0.30, 23 and 20 pptv) on 31 August. To put our simulated results in a global perspective, the $RO_X$ radical levels at TC are well within the measured or modeled ranges in the polluted urban environments (Stone et al., 2012; and references therein). For instance, the peak

concentrations of OH and $HO_2$ in Hong Kong are higher than those measured in Los Angeles U.S. (George et al., 1999), Birmingham U.K. (Emmerson et al., 2005) and Tokyo Japan (Kanaya et al., 2007), similar to those in New York U.S. (Ren et al., 2003) and Mexico City Mexico (Dusanter et al., 2009; Shirley et al., 2006), and lower than those in Nashville U.S. (Martinez et al., 2003) and Houston U.S. (Mao et al., 2010). In comparison with the limited results available in China, the $RO_X$ levels at TC are

comparable to those simulated in Beijing (Liu et al., 2012), Mt. Tai (Kanaya et al., 2009) and the PRD region (Hofzumahaus et al., 2009), but are much lower than the in-situ observations at a rural site in the PRD which cannot be explained by the currently known OH sources (Hofzumahaus et al., 2009; Lu et al., 2012). In the following section, a detailed budget analysis of the radical initiation, recycling and termination processes is presented.

### 3.3.  $RO_X$ budget analysis


*Figure 6* presents the primary daytime sources of OH, $HO_2$ and $RO_2$ at TC on 25 August 2011, and the detailed daytime $RO_X$ budget is schematically illustrated in *Fig. 7* (note that the results on 31 August are similar and shown in *Figs. S5 and S6*). HONO photolysis is not only the predominant source of OH in the early morning but also a major source throughout the daytime. Photolysis of $O_3$ becomes an

5 important OH source at midday, the strength of which is comparable to that of HONO photolysis. In terms of the daytime average (6:00–18:00, LT), HONO photolysis is the dominant OH source with an average OH production rate of 1.5 ppbv/h, followed by $O_3$ photolysis (0.9 ppbv/h). In addition, ozonolysis reactions of unsaturated VOCs are another considerable OH source with a mean production rate of 0.2 ppbv/h, whilst other sources (*e.g.,* photolysis of $H_2O_2$, $HNO_3$ and OVOCs) are generally

negligible.

For $HO_2$, the most important source is the photolysis of OVOCs with a daytime average production rate of 2.7 ppbv/h. Specifically, photolysis of formaldehyde produces $HO_2$ at a rate of 0.8 ppbv/h, while the remaining majority (1.9 ppbv/h) is from the photolysis of the other OVOCs. In addition, another source that needs to be considered is reactions of $O_3$ with unsaturated VOCs, which produce $HO_2$ at 0.1

15 ppbv/h on average during the daytime.

As to $RO_2$, photolysis of OVOCs presents the dominant source with a daytime mean production rate of 1.9 ppbv/h. The $NO_3$ oxidation of VOCs is the second most significant $RO_2$ source at TC, contributing 0.5 ppbv/h of daytime $RO_2$ production. This result suggests that $NO_3$ may play an important role in the daytime chemistry of the polluted atmosphere, and is different from most results

obtained elsewhere which have indicated the negligible role of $NO_3$ in the daytime photochemistry (a detailed analysis is presented in Section 3.4). Ozonolysis reactions of VOCs also contribute moderately to the daytime $RO_2$ production (0.2 ppbv/h). Furthermore, oxidation of VOCs by the chlorine atoms, which are produced by photolysis of the nocturnally formed $ClNO_2$, is another $RO_2$ source (0.1 ppbv/h), particularly in the early morning period (with a maximum of 0.4 ppbv/h).

From the $RO_X$ perspective, the primary radical production in Hong Kong is dominated by photolysis of OVOCs (except for HCHO), followed by photolysis of HONO, $O_3$ and HCHO, and reactions of $O_3$+VOCs and $NO_3$+VOCs. Comparison of Hong Kong with other metropolitan areas clearly reveals the heterogeneity in radical chemistry in different urban environments. For example, the dominant radical sources are $O_3$ photolysis in Los Angeles (Griffin, 2004) and Nashville (Martinez et al.,

2003), HONO photolysis in New York City (Ren et al., 2003), Paris (Michoud et al., 2012) and Santiago (Elshorbany et al., 2009), HCHO photolysis in Milan (Alicke et al., 2002), and OVOC photolysis in Hong Kong, Beijing (Liu et al., 2012), Mexico City (Volkamer et al., 2010), Birmingham (summer case; Emmerson et al., 2005) and Chelmsford near London (Emmerson et al., 2007). This highlights the variability of the initiation mode of atmospheric photochemistry, which ultimately drives the formation

of ozone and secondary aerosols in urban atmospheres.



Efficient recycling of radicals can be also illustrated in *Fig. 7*. Oxidation of CO and VOCs by OH produces $HO_2$ and $RO_2$ with daytime average rates of 3.3 and 8.0 ppbv/h, respectively. Reactions of $RO_2+NO$ and $HO_2+NO$ in turn result in strong production of RO (9.0 ppbv/h) and OH (12.5 ppbv/h), with $O_3$ formed as a by-product. It is evident that these recycling processes dominate the total production of OH, $HO_2$ and $RO_2$ radicals. It is common that the radical propagation is efficient and amplifies the effect of the newly-produced radicals in the polluted atmospheres with the co-existence of abundant $NO_X$ and VOCs (Elshorbany et al., 2009; Liu et al., 2012). As to the termination processes, the $RO_X$ radical sink is clearly dominated by their reactions with $NO_X$. Specifically, reactions of $OH+NO_2$ and $RO_2+NO_2$, forming $HNO_3$ and organic nitrates, contributed approximately 2.8 and 2.5 ppbv/h of the radical loss on daytime-average at TC. This is in line with the understanding that reactions with $NO_X$ usually dominate the radical sink in high-$NO_X$ environments.

### 3.4. Evidence of daytime $NO_3$ chemistry

The $NO_3$ radical can initiate the oxidation of VOCs and lead to formation of $RO_2$ and nitrogen-containing organic aerosols (Rollins et al., 2012; Saunders et al., 2003). These processes are usually considered to mainly occur at night and be negligible during the daytime due to the fast photolysis of $NO_3$. In the present study, we observed an interesting case that provided evidence of the operation of daytime $NO_3$-initiated chemistry. The detailed measurement data of chemical and meteorological parameters in this case (*i.e.,* 25 August 2011) are depicted in *Fig. 8*. During this episode, the air was characterized by high concentrations of $O_3$ (up to 170 ppbv), $NO_2$ (~25 ppbv as the afternoon average) and VOCs (see *Table 1*). Meanwhile, the solar irradiation arriving at the surface was weaker than other days, as evidenced by the relatively lower values of $J_{NO2}$ (with a peak of $6.0\times10^{-3}$ s$^{-1}$) compared to clear days with $>10\times10^{-3}$ s$^{-1}$ (see *Fig. S2*). The ambient relative humidity (RH) in the afternoon was in the range of 60%-70%, indicative of the 'cloud-free' conditions, whilst the aerosol scattering coefficient was very high (up to 525 Mm$^{-1}$; compared to $28\pm12$ Mm$^{-1}$ on clear days). Hence, the attenuated solar radiation is attributed to the abundant aerosol loadings. Under such conditions, the model produced an afternoon peak of $NO_3$ of ~7 pptv at 13:30–15:00 LT (except for the maximum of 11.3 pptv at 14:50 LT that was coincident with an extremely low solar radiation condition).

To further understand the causes and impacts of the daytime $NO_3$ chemistry, a detailed budget analysis was conducted with the OBM-AOCP model. The midday average (9:00 – 15:00 LT) production and destruction rates of $NO_3$ from the individual reaction pathways are documented in *Fig. 9*. The co-existence of high concentrations of $O_3$ and $NO_2$ resulted in a very strong $NO_3$ production with an average strength of 11.0 ppb/h. Given its high reactivity, $NO_3$ once formed, can be readily photolysed as well as react with NO and VOCs. For this case, about 80% (*i.e.,* 8.8 ppb/h) of $NO_3$ reacted with NO to convert back to $NO_2$. Due to the weak solar radiation, photolysis only accounted for 6.2% (or 0.7 ppb/h) of the $NO_3$ loss. In comparison, reactions of $NO_3$ with VOCs contributed 11.7% (or 1.3 ppb/h) to the





total loss at midday. During this episode, therefore, $NO_3$ appeared to be the second most important oxidant (see *Section 3.2*) and the reactions of $NO_3$ with VOCs presented a considerable $RO_2$ source during the daytime (*Section 3.3*). In addition, the $NO_3$-initiated degradation of VOCs could also lead to formation of secondary organic nitrate aerosols, but was not simulated in the present study.

The above analysis indicates the possible importance of $NO_3$-initiated oxidation in the daytime atmospheric photochemistry under specific conditions. This analysis is solely derived from an observation-based modeling study of a unique pollution case in Hong Kong. Nonetheless, we hypothesize that it may also take place in other polluted urban atmospheres, especially in the large cities of China. It is known that eastern China now suffers from widespread and severe photochemical smog

during the summer, which features by elevated concentrations of $O_3$, $NO_X$, VOCs, and fine particulate matter (Xue et al., 2014a). The intense air pollution usually induces 'smoldering' weather with poor visibility and hence attenuated solar irradiation (Ding et al., 2013). All these unfavorable conditions would facilitate the operation of daytime $NO_3$ chemistry as found in Hong Kong in the present study. Further studies are required to verify this phenomenon in other polluted environments and quantify its

contributions to the formation of ozone and secondary organic aerosols.

### 3.5. Sensitivity of radical sources to precursors

    As photochemical $O_3$ and aerosol formation is essentially limited by the availability of radicals, it is vitally important to examine the sensitivity of primary radical production to photochemical precursors, the emission of which can be directly controlled, i.e. $NO_X$ and VOCs. For this purpose, a series of

sensitivity model runs were performed with reduced concentrations of $NO_X$ or individual groups of VOCs. Given the fact that $NO_X$ and VOCs affect $RO_X$ production largely through degradation of their reaction intermediates and products (*i.e.,* $O_3$, HONO and OVOCs), here the model was not constrained by the measured data of $O_3$, HONO and OVOCs. The relative changes in the primary $RO_X$ production rates with compared to without reduced precursor concentrations can be considered as the impacts of

the target precursor on radical sources.

    The impacts of $NO_X$ and different VOC groups on the primary production of OH, $HO_2$, $RO_2$, and the sum of $RO_X$ on 25 August 2011 are illustrated in *Fig. 10*. It can be clearly seen that the radical sources, in particular for $HO_2$ and $RO_2$, are most sensitive to the aromatic VOCs at TC. Controlling aromatics would suppress the formation of $O_3$ (the $O_3$ formation is aromatics-limited; figures not shown)

and OVOCs and in turn reduce the primary production of $RO_X$ radicals, and vice versa. In comparison, the $RO_X$ sources are moderately sensitive to alkenes and insensitive to some degree to alkanes. Reducing $NO_X$ would, as expected, lead to a decreased supply of OH, and this should be due to the attenuated heterogeneous formation of HONO, a major OH precursor at TC. In contrast, reducing $NO_X$ would also result in an increase in $O_3$ (note that the $O_3$ formation is in a $NO_X$-titrated regime) and

OVOCs levels, which can enhance the production of $HO_2$ and $RO_2$ radicals. Combining the results



indicates that $NO_X$ plays an overall negative role in the primary $RO_X$ sources in Hong Kong. Consequently, controlling aromatics is the most efficient way to reduce the atmospheric oxidative capacity and mitigate the photochemical formation of ozone and secondary aerosols in Hong Kong.

## 4. Conclusions

The detailed atmospheric photochemistry during a severe smog episode in Hong Kong is analyzed. A strong oxidative capacity of the atmosphere is found and ascribed to OH and to a lesser extent $NO_3$. Elevated concentrations of $O_3$, $NO_2$, HONO and VOCs were concurrently observed, which resulted in the strong production of $RO_X$ and $NO_3$ as well as efficient radical recycling. Photolysis of OVOCs other than HCHO was found to be the dominant primary $RO_X$ source, followed by photolysis of HONO, $O_3$

and HCHO, and reactions of $O_3$+VOCs and $NO_3$+VOCs. Reducing aromatic hydrocarbons would efficiently suppress the primary production of $RO_X$, and hence lessen the oxidative capacity of the atmosphere in Hong Kong. Controlling $NO_X$ is effective for reducing the supply of OH, but would increase more significantly the primary production of $HO_2$ and $RO_2$.

On 25 August 2011, a unique case when heavy air pollution attenuated the solar irradiation reaching

the surface in Hong Kong, $NO_3$ was identified as an important oxidant in the daytime chemistry. VOC oxidation by $NO_3$ represented the second largest source of $RO_2$, with a daytime average production rate of 0.5 ppbv/h. The $NO_3$-initiated degradation of VOCs would enhance the formation of $O_3$ and nitrogen-containing organic aerosols. This study indicates the potential operation of the daytime $NO_3$ chemistry in polluted urban atmospheres characterized by the co-existence of abundant $O_3$, $NO_2$, VOCs

and particles. Further studies, especially direct observations of $NO_3$, are required to verify this interesting phenomenon in other environments and to evaluate its contribution to the $O_3$ and secondary organic aerosol formation.

## Acknowledgements

The authors appreciate Steven Poon, Yee Jun Tham, Shengzhen Zhou, Wei Nie and Jia Guo for

their contributions to the field study; the University of Leeds for providing the Master Chemical Mechanism; and the NOAA Air Resources Laboratory for providing the web-based HYSPLIT model. The field observations were funded by the Environment and Conservation Fund of Hong Kong (project No.: 7/2009), and the data analyses were supported by the National Natural Science Foundation of China (project No.: 41505111) and Qilu Youth Talent Programme of Shandong University.

**Disclaimer**

The opinions expressed in this paper are those of the authors and do not necessarily reflect the views or policies of the Government of the Hong Kong Special Administrative Region, nor does mention of trade names or commercial products constitute an endorsement or recommendation of their use.



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



**Table 1.** 24-hour average concentrations of hydrocarbons and oxygenated VOCs measured at Tung Chung on 25 and 31 August 2011.[a]

| Species | 25 August | 31 August | Species | 25 August | 31 August |
|---|---|---|---|---|---|
| methane | 2.264 | 2.275 | benzene | 1008 | 569 |
| ethane | 1192 | 525 | toluene | 9465 | 3557 |
| propane | 2717 | 1589 | ethylbenzene | 1718 | 700 |
| *n*-butane | 3751 | 1361 | *o*-xylene | 979 | 328 |
| *i*-butane | 2614 | 929 | *m*-xylene | 2082 | 935 |
| *n*-pentane | 1175 | 561 | *p*-xylene | 813 | 239 |
| *i*-pentane | 1569 | 817 | propylbenzene | 63 | 24 |
| *n*-hexane | 1161 | 1039 | *i*-propylbenzene | 54 | 20 |
| *n*-heptane | 519 | 297 | *o*-ethyltoluene | 140 | 82 |
| *n*-octane | 150 | 410 | *m*-ethyltoluene | 338 | 167 |
| *n*-nonane | 133 | - | *p*-ethyltoluene | 143 | 113 |
| *2*-methylpentane | 1123 | - | *1,2,3*-trimethylbenzene | 204 | 46 |
| *3*-methylpentane | 842 | - | *1,2,4*-trimethylbenzene | 515 | 338 |
| ethene | 1861 | 681 | *1,3,5*-trimethylbenzene | 124 | 49 |
| propene | 537 | 482 | formaldehyde | 9890 | 8968 |
| *1*-butene | 196 | 136 | acetaldehyde | 4250 | 3990 |
| *i*-butene | 224 | 282 | propanal | 940 | 670 |
| *trans-2*-butene | 68 | 36 | acetone | 590 | 10670 |
| *cis-2*-butene | 54 | 23 | butanal | 640 | 269 |
| *1,3*-butadiene | 72 | 57 | pentanal | 1420 | 1596 |
| *1*-pentene | 50 | 16 | hexanal | 200 | 506 |
| Isoprene | 779 | 65 | benzaldehyde | 890 | 660 |
| *α*-pinene | 92 | 48 | methyl ethyl ketone | 260 | 1027 |
| *β*-pinene | 36 | 21 | acrolein | 30 | BDL |
| ethyne | 2903 | 265 | crotonaldehyde | 30 | 510 |

[a] The units are pptv except for methane which is in ppmv. "-" indicates no data available, and "BDL" indicates below detection limit.



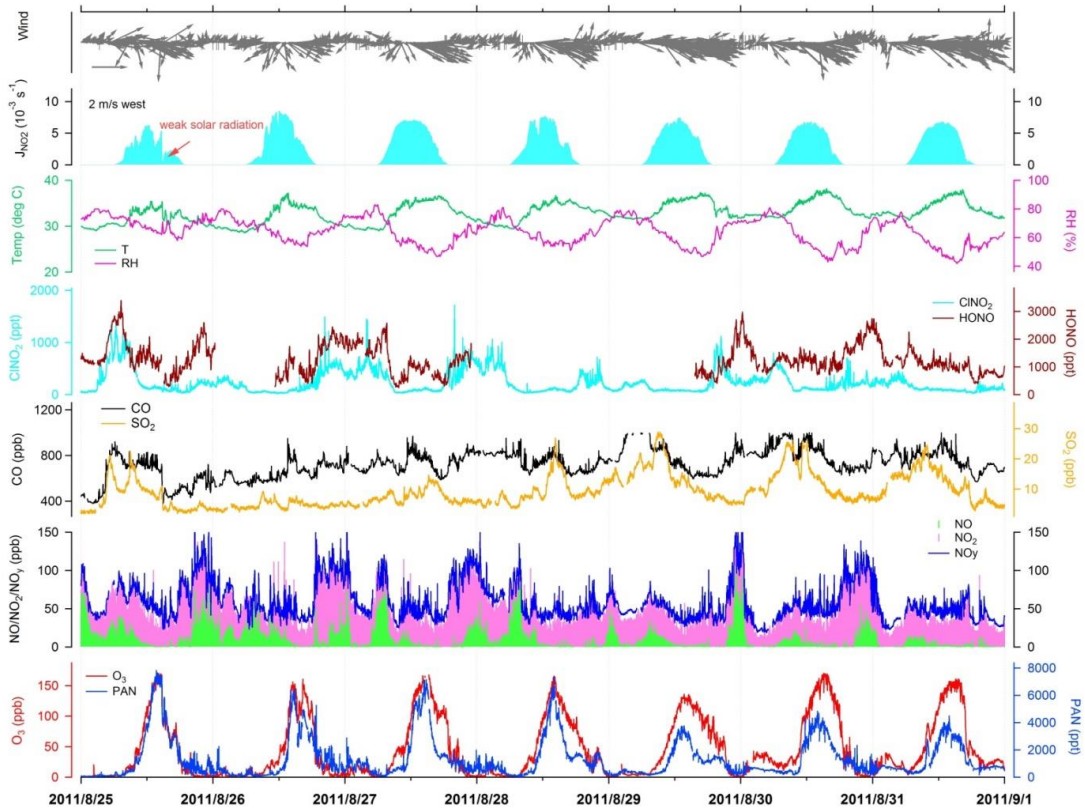

**Figure 1.** Time series of trace gases and meteorological parameters observed at Tung Chung from 25-31 August 2011. The data gap for HONO was mainly due to the calibration and maintenance of the instruments.





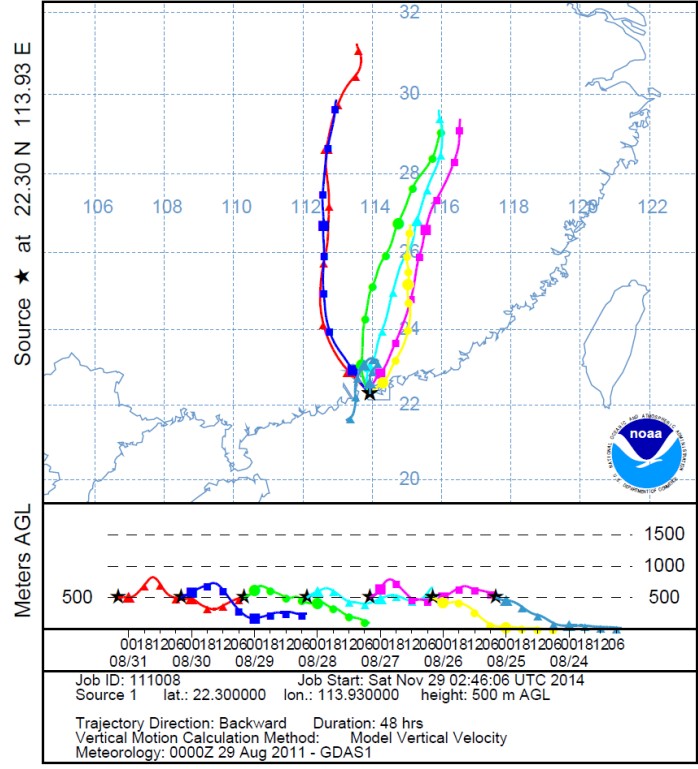

**Figure 2.** 48-hour backward trajectories calculated by the HYSPLIT model for air masses at Tung Chung at 12:00 LT each day from 25 – 31 August 2011. The height of the starting endpoint was set as 500 m a.g.l. in the model.




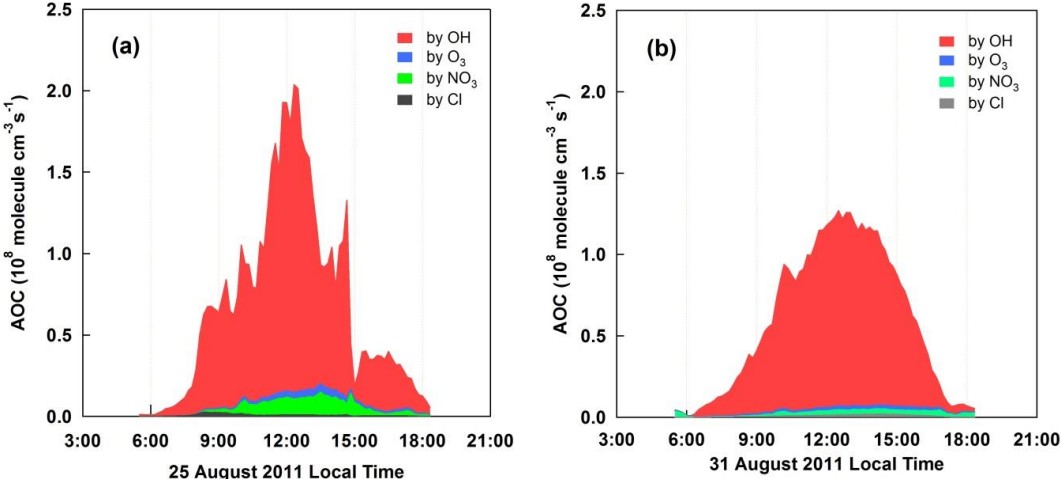

**Figure 3.** Daytime atmospheric oxidative capacity (AOC) and contributions of major oxidants at Tung Chung on (a) 25 August 2011 and (b) 31 August 2011

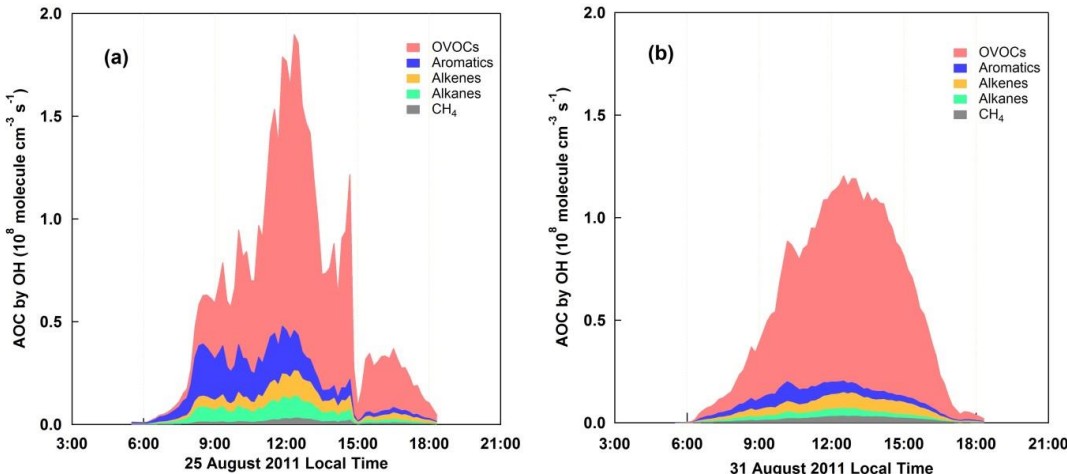

**Figure 4.** Partitioning of the OH-contributed AOC (or OH reactivity) to the oxidation of major VOC groups at Tung Chung on (a) 25 August 2011 and (b) 31 August 2011





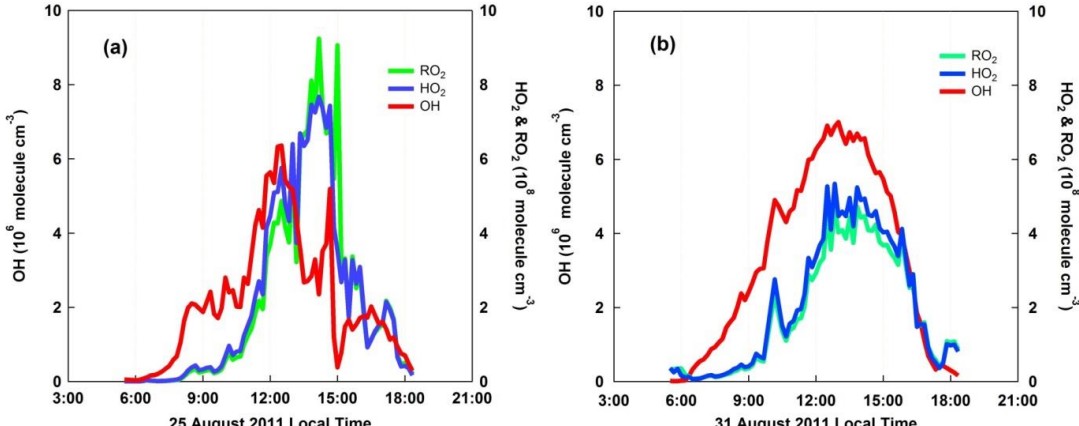

**Figure 5.** Model-simulated daytime concentrations of OH, HO$_2$ and RO$_2$ radicals at Tung Chung on (a) 25 August 2011 and (b) 31 August 2011



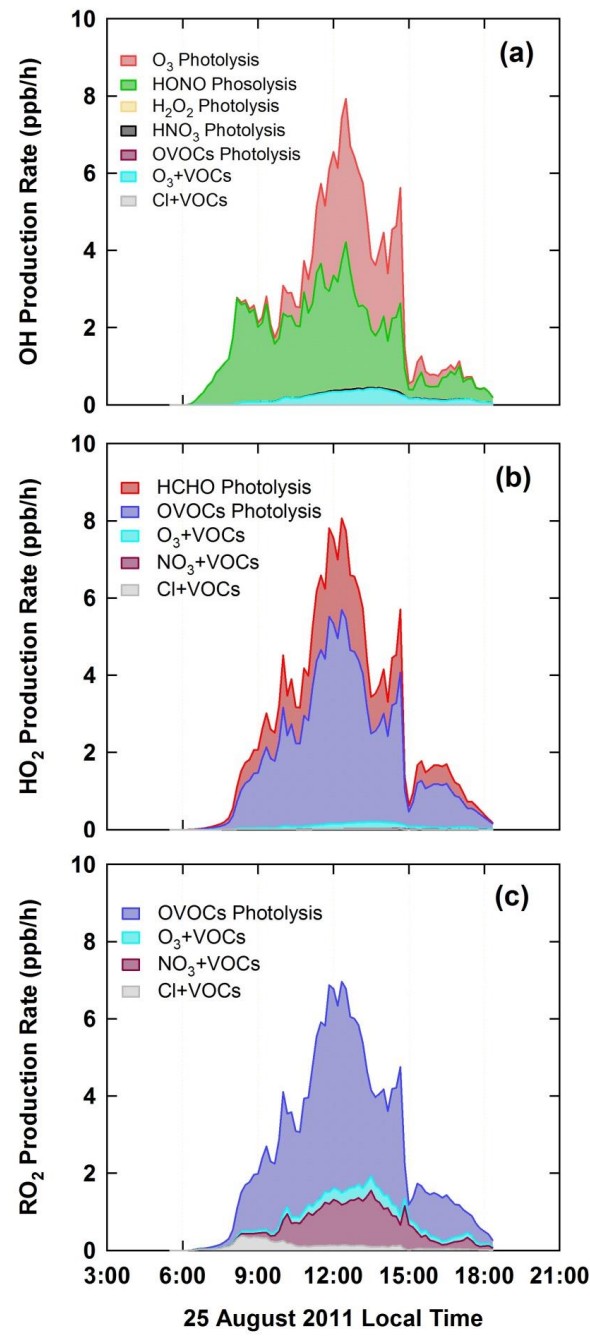

**Figure 6.** Primary daytime sources of (a) OH, (b) HO$_2$ and (c) RO$_2$ radicals at Tung Chung on 25 August 2011



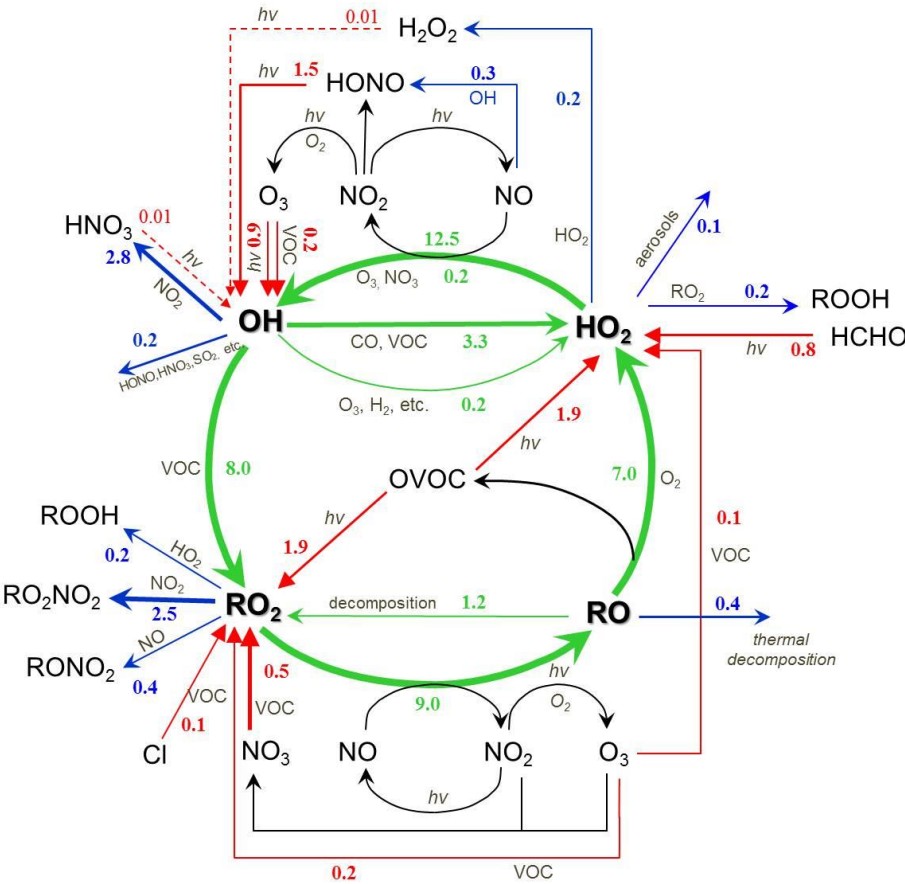

**Figure 7.** Daytime average RO$_X$ budget at Tung Chung on 25 August 2011. The unit is ppb/h. The red, blue and green lines indicate the production, destruction and recycling pathways of radicals, respectively.





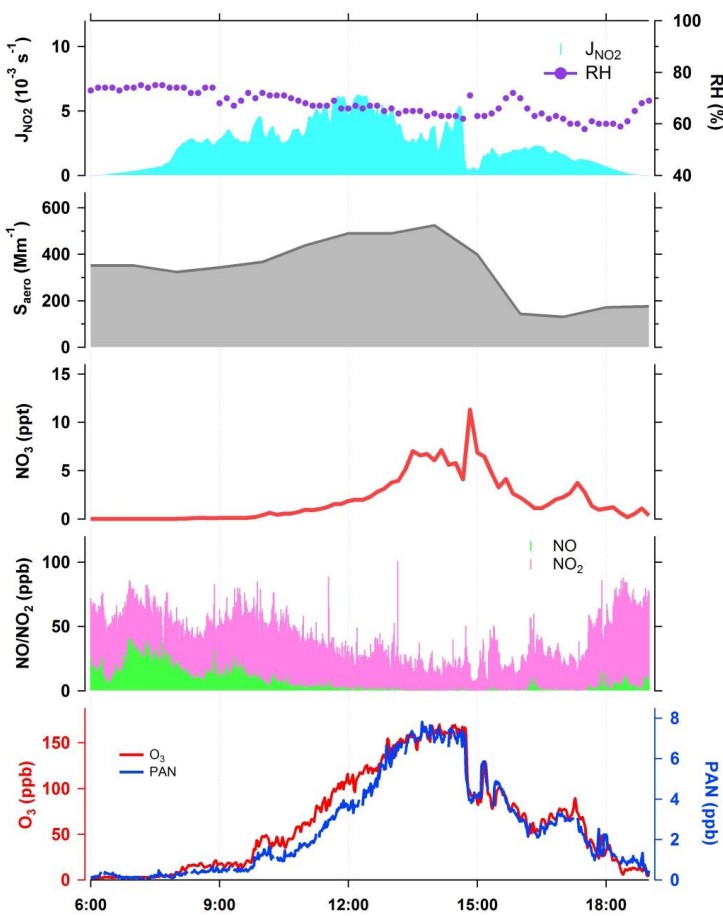

**Figure 8.** Time series of chemical and meteorological parameters observed at Tung Chung on 25 August 2011.

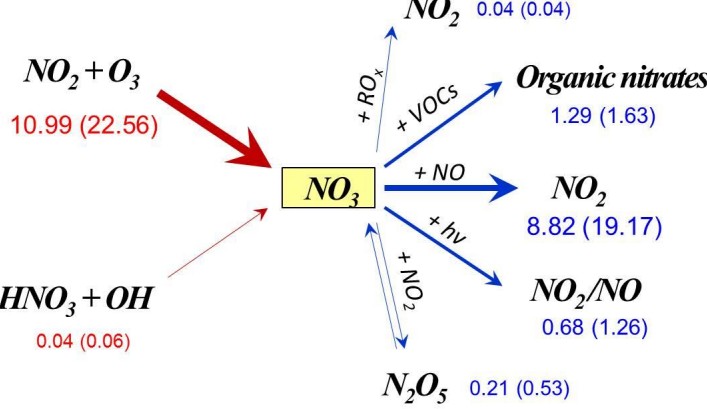

**Figure 9.** Midday average (9:00–15:00 local time) budget of the $NO_3$ radical at Tung Chung on 25 August 2011. The units are ppb/h. Peak values are also given in parentheses.





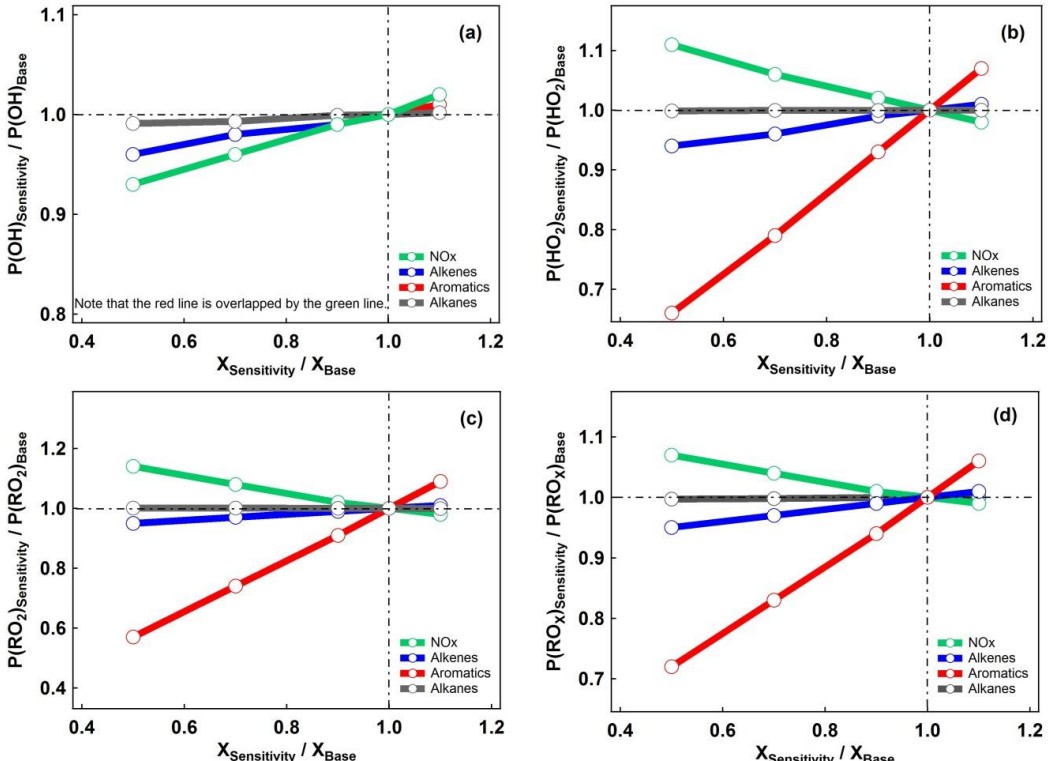

**Figure 10.** Sensitivity of primary production rates of (a) OH, (b) HO$_2$, (c) RO$_2$ and (d) RO$_X$ to NO$_X$ and individual VOC groups at Tung Chung on 25 August 2011.