# Peer review of "Oxidative capacity and radical chemistry in the polluted atmosphere of Hong Kong and Pearl River Delta region: analysis of a severe photochemical smog episode"

_Atmospheric Chemistry and Physics, 2016_

## Referee Comment (RC1) · Anonymous Referee #1 · 14 Mar 2016

This manuscript presents a modelling case study of the oxidative capacity of Hong Kong and the Pearl River Delta region of China during a photochemical smog episode. The dominant $RO_x$ radical sources and OH sinks are identified. $NO_3$ is identified as a non-negligible $RO_x$ source during the daytime when solar irradiation was attenuated by high aerosol loadings. This is an interesting result, and this source of $RO_x$ has not been considered to be important in earlier studies. Through sensitivity analyses, a reduction in the loading of aromatics is highlighted as an efficient way to mitigate photochemical pollution. This paper demonstrates nicely the major species and reaction pathways which control the oxidation chemistry in this region, however, there is a lack of synthesis of the results and discussion on the impact of these findings, and how these findings compare to previous urban studies is limited. In addition to a number of minor comments, I have made a number of suggestions below on where more detail is warranted or where the discussion should be developed before final publication is considered.

Pg 1, line 16: 'we analyze a multi-day photochemical smog episode..' Actually the analysis is confined to two individual days.

Pg 2, line 12: remove the space between the arrow and 'RO'.

Pg 2, lines 18 – 29: Some care is needed when discussing the relative primary radical sources at different urban sites: In some of the earlier studies HONO was not measured (e.g. Emmerson et al, 2005) and the inability of models to reproduce observed daytime HONO concentrations means that some of the differences observed may be caused by differences in model constraints. The model constraints used in these studies should be considered here along with the differences highlighted. Some comment on the relative source strengths in ppb/hr would be helpful and could be referred to when the results from the TC study are discussed.

Pg 4, line 18: Define 'MoO'.

Pg 5, line 1: Although a reference is provided which relates to the differences between the on-line and canister alkene measurements a comment should be made in this manuscript on why the canister sample are considered more reliable than the on-line measurements.

Section 2.2 The OBM-AOCP model: Further details on model parameters are required. The reader should not have to refer to the references for parameters specific to this study. What uptake coefficients were assumed for $HO_2$ and $N_2O_5$? What deposition rate was used? Which model species were deposited? Lines 28, 29: '..impacts on the modelling results were negligible': Which model species were considered when assessing the impact of mixing height?

How was $H_2$ treated in the model? Was this measured?

Are all the VOCs listed in Table 1 used as model constraints? Does this equate to 15600 reactions or does this figure refer to the number of reactions when the MCM is run in its entirety?

Pg 5, Line 33: It is unclear what the authors mean by '..and grouped into a relatively small number of major routes'. Does this grouping take place post-model run? I think this sentence needs re-wording so it is clear to the reader what analysis this relates to.

Pg 6, line 17: Change 'simulation' to 'comparison'

Pg 6, line 18: what is meant by 'reasonable estimates'? Do the authors mean until steady state conditions are reached?

Pg 6, line 26: The authors should consider replacing Figure 1 with the similar figure provided in the supplementary information (Figure S2) which highlights the contrasting conditions well and provides more information than presented in Figure 1.

Pg 6, line 30: Remove 'meanwhile'

Pg 6, line 33: '..intense photochemical oxidant production' this figure does not demonstrate this, rather, the subsequent model analysis reveals this.

Pg 7, line 3: Need to contrast these concentrations with other relevant observations, e.g. perhaps typical concentrations measured at other urban centres or could contrast with the concentrations of these species observed at TC prior to the pollution episode.

Pg 7, line 6: Elevated daytime HONO concentrations at urban locations that cannot be reproduced by models which consider only gas-phase chemistry are now well reported. So not uncommon, but interesting nonetheless.

Pg 7, line 14: Is the Ding et al. 2004 reference appropriate given that it was published before this campaign so cannot document the source of the pollution event discussed here.

Pg 7, line 20: I suggest moving figure 2 to SI. It is not critical for the paper, unlike some of the other figures currently in the SI (see later comments). Is there an order to the trajectories shown? Does Red = 25$^{th}$? It would be helpful to highlight the PRD region on the map.

Pg 7, line 22: Fig S2 does not highlight this airmass switch discussed. Should another figure be referenced here?

Pg 8, line 5: What is the % contribution of $NO_3$ during the day on the 31$^{st}$ August? Which VOC species is $NO_3$ primarily oxidising? The authors suggest that ozonolysis isn't significant here because of the 'lower abundances of alkenes' so I presume carbonyls dominate $NO_3$ reactivity? More detail needed on this.

How does figure 3 compare to similar analyses reported in the literature? E.g. Bannan et al. (JGR-Atmos, 2015)

Pg 8, line 8: Is Nitryl chloride the only source of Cl atoms in the model? What was the daytime concentration of Cl atoms predicted? Please state. How was the photolysis rate of $ClNO_2$ (and the photolysis of other Cl-containing species) incorporated into the model?

Pg 8, lines 9 and 10: poor sentence structure, please revise.

Pg 8, line 10, figure 4: For ease of comparison with earlier reported OH reactivities at urban sites, could figure 4 be presented as a breakdown of OH reactivity with units ($s^{-1}$)? i.e. divide through by modelled OH concentration.

Pg 8, lines 11 – 16: Does this partitioning include model-generated intermediate species in the OVOC segment? I note from Table 1 that elevated levels of isoprene were observed on the 25th (relative to the 31st), do the biogenics measured contribute significantly to OH reactivity at TC? How does this compare to the recent OH reactivity observations made in London (Whalley et al. ACP, 2016)?

Pg 8, lines 19 – 34: without a direct comparison to $RO_x$ observations at TC specifically this commentary adds little to the manuscript and so I suggest removing this paragraph (or expand this section to provide further discussion of the dominant radical sources and sinks in the campaigns referred to – and how these sources and sinks compare and contrast with TC).

Pg 9, line 3: I suggest Figs S5 and S6 are moved into the main paper.

Pg 9, line 13: Some information on the identity of the other OVOCs which contribute to $HO_2$ production is needed. Are these compounds which were measured directly (e.g. acetaldehyde) or oxidation products generated by the model? How do these source strengths compare to other urban locations?

Pg 9, line 16: 'As to' to 'For'

Pg 9, lines 17 – 18: Considering $RO_2$ deriving from $NO_3$+VOC reactions, this source seems to have a similar % contribution to the total $RO_2$ sources both on the 25th when $j(NO_2)$ was attenuated and also on the 31st when $j(NO_2)$ was not attenuated. A comment is needed which compares these two contrasting days.

Pg 9, line 20: '..different from most results obtained elsewhere' a reference to support this statement is needed.

Pg 9, lines 25 – 35: See earlier comment on model constraints in the various studies highlighted.

Section 3.3: I suggest that a discussion on how the 25th and 31st contrast be included at the end of this section, focussing on local versus regional influences?

Pg 10, line 23: I am not sure that RH can be used to confirm cloud-cover. Does it matter what caused this attenuation?

Pg 10, line 30, Fig. 9: Please include the modelled heterogeneous loss rate of $N_2O_5$ in this figure. What uptake of $N_2O_5$ was assumed?

Pg 11, line 1: Does the high aerosol loading impact the heterogeneous loss of $N_2O_5$?

Section 3.5: I found this section difficult to follow as the sentence structure was poor throughout. This needs a careful edit to improve the clarity.

Pg 11, lines 32 – 33: '..due to the attenuated heterogeneous formation of HONO'. Were heterogeneous sources of HONO included in the model? If they were, a description of these

sources should be included in the experimental description. If they were not included, then attenuated heterogeneous formation of HONO cannot be the cause of the reduction of OH in the model. Is this trend not simply caused by a reduction in the secondary OH source from $HO_2+NO$? In this analysis are NO and $NO_2$ concentrations both reduced by the same fraction?

Why were only alkanes, alkenes and aromatics considered in this analysis? What is the sensitivity to the other VOC groups measured, e.g. the Biogenics?

Conclusions: This section should provide some commentary on the local versus regional events; the latter sections of the paper neglect the case study on the 31[st] and it would conclude the paper nicely if the results from this day were evaluated alongside the 25[th] here.

---

## Referee Comment (RC2) · Anonymous Referee #2 · 15 Mar 2016

**A review report on "Oxidative capacity and radical chemistry in the polluted atmosphere of Hong Kong and Pearl River Delta region: analysis of a severe photochemical smog episode" by Xue et al. (2016)**

The study investigates the atmospheric radical budgets in the Hong Kong and PRD region, using a box model constrained with a full suite of ancillary measurements, not including OH or HO2. So, the budget analysis presented here is based solely on simulated OH/HO2.

My major concern is that the study does not provide new scientific results, especially given that several previous studies addressed this issue in the last decade, as also mentioned in the study. In addition, the inconsistency between simulated OH/HO2 and those previously measured in the same region weaken the results of this study and its argument toward a science-based control strategy. Add to this, several consistencies in the discussion of the results. The sensitivity analysis needs to be redone and reevaluated (see below).

The authors should address the following issues before the Manuscript can be considered for publication.

Abstract:
**Page 1, Line 29**: The statement "Sensitivity studies show that controlling aromatics is the most efficient way to reduce the atmospheric oxidative capacity and mitigate photochemical pollution in Hong Kong." does not seem correct. The atmospheric oxidation capacity is the total loss rate of all species and thus represents the atmospheric capacity to reduce/degrade the atmospheric pollutants. Thus, I think it is just irrelevant to try to reduce the atmospheric oxidation capacity.! Reducing the photochemical pollution caused by ozone and PAN, photochemical secondary products, require sensitivity analysis to determine the contribution of each VOC to ozone formation (based on their kinetic and mechanistic properties) and target these species, which often require the use of a region-specific reactivity scale.

**Page 7, line 10**: The statement "abundant VOCs would facilitate efficient radical recycling" needs revision, since the efficient recycling of peroxy radical (RO2/HO2+NO=OH) requires only reasonable amount of NO. Under VOC-sensitive conditions, higher VOC lead will lead to higher ROx productions.

**Page 7, Line 22**: I do not see the wind direction in Fig. S2?

**Page 8, lines 8**: would be also informative if the authors could compare these AOC values with other world regions from previous studies and show its significance.

**Page 8, lines 19**: Please show $j(O^1D)$ on the same figure with OH/HO2 (figure 5).

**Page 8, lines 29-33**: So, now I see some comparisons but it is not consistent with the study's results. The simulated OH/HO2 are about 2 times lower than the measured OH/HO2 at PRD (Hofzumahaus et al., 2009).

**Page 9, line 1**: Figure 6(a): Since the contributions of the photolysis of H2O2, HNO3, OVOCs seem extremely low that is not even seen on the figure, why they are shown?

The contribution of HONO photolysis in this figure should be only the net HONO (subtract [HONO]pss from OH+NO=HONO since it is not a net OH source).

**Page 9, lines 25-35**: The authors should discuss if the mentioned heterogeneity in the primary sources is just a result of not measuring all sources (i.e, HONO was not measured in all these mentioned studies) rather than differences in regional source contributions.

**Page 11, lines 22-25**: Why measured HONO is not constrained. The simulated HONO by the model represents only the [HONO]pss (OH+NO=HONO, HONO+hv=OH+NO) and does not represent a net source of radicals. [HONO]pss is a direct gas phase reaction of OH+NO and is not as secondary oxidation product. Not including measured HONO will certainly underestimate the simulated OH and thus will affect the simulated secondary products (e.g., O3).

**Page 11, line 32**: The discussion in this paragraph is not clear, how decreasing NO2 would decrease OH? Decreasing the OH loss via OH+NO2 reaction (via decreasing NO2) should lead to increased OH. Decreasing NO (only NO) would decrease OH (via decreasing the reaction rate of RO2/HO2+NO=OH).

**Page 11, line 33**: The authors did not mention before if they included a mechanism for heterogeneous formation of HONO? How this was considered parallel to measured HONO?
        How reducing NOx would decrease OH (line 32) and increase O3 (line 34)?

**Page 11, line 34**: What is a "NOx-titrated regime", its not defined anywhere in the text?

Minor comments:

Could the authors address the differences between the MCMv3.2 and the most recent version, and how this would affect their analysis?

---

## Author Comment (AC1) · 23 May 2016

**Response to Referee 2**

*A review report on "Oxidative capacity and radical chemistry in the polluted atmosphere of Hong Kong and Pearl River Delta region: analysis of a severe photochemical smog episode" by Xue et al. (2016)*

*The study investigates the atmospheric radical budgets in the Hong Kong and PRD region, using a box model constrained with a full suite of ancillary measurements, not including OH or $HO_2$. So, the budget analysis presented here is based solely on simulated $OH/HO_2$.*

*My major concern is that the study does not provide new scientific results, especially given that several previous studies addressed this issue in the last decade, as also mentioned in the study. In addition, the inconsistency between simulated $OH/HO_2$ and those previously measured in the same region weaken the results of this study and its argument toward a science-based control strategy. Add to this, several consistencies in the discussion of the results. The sensitivity analysis needs to be redone and reevaluated (see below).*

*The authors should address the following issues before the Manuscript can be considered for publication.*

Response: we thank the reviewer very much for the critical comments which would definitely help us to improve our work. The major concerns of the reviewer are on (1) the significance of the scientific results given lack of $HO_x$ observations and (2) the reliability of the sensitivity analyses. Below we first address these major concerns and then reply individually the specific comments. For clarity, the reviewer's comments are listed below in black italics, while our responses and changes in manuscript are shown in blue and red, respectively.

**(1) On the significance of this study**

We agree with the reviewer that the present study is only based on the simulation of $RO_X$ radicals with a full suite of ancillary observations. Without direct $RO_X$ measurements (which are still not available so far in Hong Kong), it is difficult to address the potential 'missing' recycling pathways of $HO_X$ radicals. Here we just would like to state the **rationale** of this study, which is to **identify the major species and reaction pathways controlling radical**

**chemistry based on the 'Known Chemistry' as well as comprehensive measurements** of related species/parameters. We think these results should be helpful for better understanding the atmospheric oxidation chemistry in the high-$NO_X$ environment of Hong Kong.

Indeed, several interesting studies have demonstrated higher than can be predicted levels of $HO_X$ at two rural sites (somewhat with low-$NO_X$) in the PRD region. To our knowledge, the present study appears to be the first effort to comprehensively quantify the radical budget in Hong Kong, which is generally featured by the high-$NO_X$ condition. Previous studies have suggested that current models are usually capable of predicting the measured $HO_X$ under the high-$NO_X$ condition. Hence **our study may add some new information about the radical chemistry (*e.g.,* major radical sources) in the high-$NO_X$ environment of the region**.

Moreover, an interesting result of this study is **the potential role of $NO_3$ in the daytime under certain conditions**. We found in one case that $NO_3$-initiated oxidation of VOCs was a considerable $RO_X$ source when the solar radiation was attenuated, possibly by high aerosol pollution. This source has not been considered to be important in earlier studies. This result suggests the possible impact of daytime $NO_3$ oxidation in the polluted atmospheres under conditions with co-existence of abundant $O_3$, $NO_2$, VOCs and particles, which is common in the metropolitan areas and fast developing regions (*e.g.,* China).

Overall, although the present study doesn't address the 'discrepancy' between observed and modeled $HO_X$ concentrations as found in other areas of the PRD region, it provides some new insights into the radical ($RO_X$ and $NO_3$) chemistry, i.e., major primary sources of $RO_X$ and potential role of $NO_3$ in daytime chemistry, in the high-$NO_X$ environment of Hong Kong. These results should be useful for the community to understand the atmospheric chemistry in different metropolitan areas of the world.

**(2) On the reliability of the sensitivity analyses**

The rationale of the sensitivity analyses is to examine the sensitivity of primary radical production (**not the concentrations**) to the controllable precursors (e.g., $NO_X$ and VOCs). We agree with the reviewer that it is irrelevant to try to reduce the atmospheric oxidation capacity. Thus the sensitivity analyses have been removed from the revised manuscript.

*Abstract:*

***Page 1, Line 29****: The statement "Sensitivity studies show that controlling aromatics is the most efficient way to reduce the atmospheric oxidative capacity and mitigate photochemical pollution in Hong Kong." does not seem correct. The atmospheric oxidation capacity is the total loss rate of all species and thus represents the atmospheric capacity to reduce/degrade the atmospheric pollutants. Thus, I think it is just irrelevant to try to reduce the atmospheric oxidation capacity.! Reducing the photochemical pollution caused by ozone and PAN, photochemical secondary products, require sensitivity analysis to determine the contribution of each VOC to ozone formation (based on their kinetic and mechanistic properties) and target these species, which often require the use of a region‑specific reactivity scale.*

Response: we agree the point of the reviewer that it is irrelevant to try to reduce the atmospheric oxidation capacity. This sentence and sensitivity analyses (Section 3.5) have been deleted from the revised manuscript.

***Page 7, line 10****: The statement "abundant VOCs would facilitate efficient radical recycling" needs revision, since the efficient recycling of peroxy radical ($RO_2/HO_2+NO=OH$) requires only reasonable amount of NO. Under VOC-sensitive conditions, higher VOC will lead to higher ROx productions.*

Response: agree. This statement has been revised as follows.

"High abundances of $O_3$, HONO and carbonyls would definitely lead to strong production of $RO_X$ radicals, and the abundant VOCs would facilitate efficient radical propagation (e.g., $OH{\rightarrow}RO_2$)."

***Page 7, Line 22****: I do not see the wind direction in Fig. S2?*

Response: wind sectors have been plotted in the revised figure. Note that Figure S2 has been moved to the main manuscript by replacing the original Figure 1. To clearly show the airmass switch, moreover, back trajectories have been shown day by day throughout the measurement period in the supplementary materials of the revised paper.

***Page 8, lines 8****: would be also informative if the authors could compare these AOC values*

*with other world regions from previous studies and show its significance.*

Response: the literatures about such kind of modeling AOC analysis are not too much. We compared our results with the available previous studies. The AOC values in Hong Kong were much higher than those determined at a rural site in Germany (e.g., 24-h average of $2.6 \times 10^6$ molecules $cm^{-3}$ $s^{-1}$; Geyer et al., 2001), but lower than that determined at a highly polluted site of Santiago, Chile (e.g., maximum of $3.2 \times 10^8$ molecules $cm^{-3}$ $s^{-1}$; Elshorbany et al., 2009). The following statement has been added in the revised manuscript.

"Such levels of AOC at TC are much higher than those determined from a rural site in Germany (Geyer et al., 2001), but a bit lower than that assessed from a polluted area in Santiago, Chile (Elshorbany et al., 2009)."

Geyer A., Alicke B., Konrad S., Schmitz T., Stutz J., Platt U.: Chemistry and oxidation capacity of the nitrate radical in the continental boundary layer near Berlin, *J Geophys. Res.,* 106, 8013-8025, 2001.

Elshorbany, Y. F., Kurtenbach, R., Wiesen, P., Lissi, E., Rubio, M., Villena, G., Gramsch, E., Rickard, A. R., Pilling, M. J., and Kleffmann, J.: Oxidation capacity of the city air of Santiago, Chile, Atmos Chem Phys, 9, 2257-2273, 2009.

***Page 8, lines 19****: Please show j(O1D) on the same figure with OH/HO₂ (figure 5).*

Response: in the present study, $J_{(O1D)}$ was not in-situ measured and was only scaled with the measured $J_{NO2}$ in the model. In the revised manuscript, we have plotted the measured $J_{NO2}$ values along with OH and $HO_2$ in Figure 5 (note that Figure 5 has been moved to the SI).

***Page 8, lines 29‒33****: So, now I see some comparisons but it is not consistent with the study's results. The simulated OH/HO₂ are about 2 times lower than the measured OH/HO₂ at PRD (Hofzumahaus et al., 2009).*

Response: yes, our simulated concentrations of OH/$HO_2$ are much lower than the measured levels at a rural site in the northern PRD. The discrepancy may be due to the difference in the sites/environments (e.g., high-$NO_X$ condition at TC and somewhat low-$NO_X$ condition at BG) and/or the deficiency of current models to understand the radical chemistry (e.g., the missing recycling pathways of $HO_X$ radicals). Without direct observations of $RO_X$, it is impossible to

address such discrepancy between measured and modeled radical levels, which is usually found in the low-$NO_X$ environments. As stated above in the response to major concerns, the rationale of this study is to identify the major species and reaction pathways affecting the radical chemistry in the high-$NO_X$ environment of Hong Kong and PRD region, based on the 'known chemistry' as well as comprehensive measurements of related species and parameters. We have stated the rationale and limitation of our study in the revised manuscript. Direct measurements of $RO_X$ radicals are quite needed to better understand the potential 'missing' pathways of radical chemistry.

*Page 9, line 1: Figure 6(a): Since the contributions of the photolysis of $H_2O_2$, $HNO_3$, OVOCs seem extremely low that is not even seen on the figure, why they are shown?*

Response: the contributions of photolysis of $H_2O_2$, $HNO_3$ and OVOCs have been removed from the revised figure, and only the major sources are shown now.

*The contribution of HONO photolysis in this figure should be only the net HONO (subtract [HONO]pss from OH+NO=HONO since it is not a net OH source).*

Response: this figure has been revised as suggested by only showing the contribution of net HONO (subtracting [HONO]pss).

*Page 9, lines 25-35: The authors should discuss if the mentioned heterogeneity in the primary sources is just a result of not measuring all sources (i.e, HONO was not measured in all these mentioned studies) rather than differences in regional source contributions.*

Response: we have reviewed these previous studies again. HONO was measured in most of these studies by various techniques including LOPAP, LP-DOAS and wet chemistry method. In three of these earlier efforts, i.e., Griffin et al. (2004) and Emmerson et al. (2005 and 2007), HONO was not measured and was only simulated with a chemical box model. In the revised manuscript, we have deleted the old reference of Griffin et al. (2004; their measurements were conducted in 1993), and added the following statement to clarify the difference in the observations.

"It is worth noting that HONO was not measured at Birmingham and Chelmsford but only

simulated by a chemical box model, and thus the contributions of HONO photolysis were likely underestimated."

***Page 11, lines 22-25****: Why measured HONO is not constrained. The simulated HONO by the model represents only the [HONO]pss (OH+NO=HONO, HONO+hv=OH+NO) and does not represent a net source of radicals. [HONO]pss is a direct gas phase reaction of OH+NO and is not as secondary oxidation product. Not including measured HONO will certainly underestimate the simulated OH and thus will affect the simulated secondary products (e.g., $O_3$).*

Response: we are sorry that the original description is not clear enough. Our model took into account the heterogeneous formation of HONO from reactions of $NO_2$ on ground and aerosol surfaces. To assess the impacts of $NO_2$ (partly through heterogeneous formation of HONO) on primary radical production, the measured HONO was not constrained in the sensitivity model runs. Anyway, **the sensitivity analyses have been deleted in the revised manuscript** as we agree with the reviewer that it is irrelevant to try to reduce the atmospheric oxidative capacity.

***Page 11, line 32****: The discussion in this paragraph is not clear, how decreasing $NO_2$ would decrease OH? Decreasing the OH loss via OH+$NO_2$ reaction (via decreasing $NO_2$) should lead to increased OH. Decreasing NO (only NO) would decrease OH (via decreasing the reaction rate of $RO_2$/$HO_2$+NO=OH).*

Response: we are sorry that this section is not clear. By the sensitivity studies, we focused on the primary production of radicals, NOT the concentrations. We meant that decreasing $NO_2$ would decrease the primary production of OH by decreasing the heterogeneous formation of HONO. Anyway, **the sensitivity analysis (Section 3.5) has been deleted in the revised manuscript**, see above.

***Page 11, line 33****: The authors did not mention before if they included a mechanism for heterogeneous formation of HONO? How this was considered parallel to measured HONO?*

Response: we are sorry that we didn't clearly state this, which made the manuscript confusing. Our model includes the heterogeneous formation of HONO from reactions of $NO_2$ on ground

and aerosol surfaces. In the revised manuscript, a detailed description of the model has been provided in the supplementary materials. Again, this section has been deleted in the revision.

*How reducing NOx would decrease OH (line 32) and increase $O_3$ (line 34)?*

Response: as stated above, reducing $NO_X$ would decrease the primary OH production (NOT the concentration) via decreasing the heterogeneous formation of HONO. According to our sensitivity studies, the $O_3$ production at TC is highly VOC-limited and in a $NO_X$-saturated regime. Reducing $NO_X$ would lead to increased $O_3$ by weakening the NO titration. Again, this section has been removed from the revised manuscript.

***Page 11, line 34****: What is a "NOx-titrated regime", it's not defined anywhere in the text?*

Response: as stated above, this discussion has been removed from the revised version.

*Minor comments:*

*Could the authors address the differences between the MCMv3.2 and the most recent version, and how this would affect their analysis?*

Response: the updates of MCM *v3.3.1* against the MCM *v3.2* are mainly on the chemistry of biogenic VOCs, including the degradation of isoprene and ozonolysis rate constants of α/β-pinenes, limonene, and β-caryophyllene. As our site (TC) is primarily influenced by the anthropogenic pollution, and the levels of BVOCs are indeed much lower than the AVOCs. Thus the impact of different versions of mechanism should be small on the analyses in the present study.

We have rerun the model with the MCM v3.3.1, and examined the difference in the simulated primary OH production rates between both mechanisms (note that we don't conduct the same analyses as the present study with the latest version of MCM, as it is really a huge work to track more than 15000 reactions in the model within a short period). As shown in the figures below, the differences between both versions of MCM are quite small.

[Figure]

Figure 1. The modeled primary production rates of OH at TC on 25[th] August 2011 with the MCM *v3.3.1* (top panel) and the MCM *v3.2* (bottom panel). **Note that the legends and scales of y-axis are different for both plots**.

---

## Author Comment (AC2) · 23 May 2016

**Response to referee's comments**

*This manuscript presents a modelling case study of the oxidative capacity of Hong Kong and the Pearl River Delta region of China during a photochemical smog episode. The dominant ROx radical sources and OH sinks are identified. $NO_3$ is identified as a non-negligible ROx source during the daytime when solar irradiation was attenuated by high aerosol loadings. This is an interesting result, and this source of ROx has not been considered to be important in earlier studies. Through sensitivity analyses, a reduction in the loading of aromatics is highlighted as an efficient way to mitigate photochemical pollution. This paper demonstrates nicely the major species and reaction pathways which control the oxidation chemistry in this region, however, there is a lack of synthesis of the results and discussion on the impact of these findings, and how these findings compare to previous urban studies is limited. In addition to a number of minor comments, I have made a number of suggestions below on where more detail is warranted or where the discussion should be developed before final publication is considered.*

Response: we appreciate the reviewer for the positive comments and helpful suggestions. In the revised manuscript, we have addressed all of the comments, and particularly adopted the suggestion to synthesize our results and compare against existing findings of previous studies. The manuscript has been significantly revised and improved based on these suggestions. For clarity, the reviewer's comments are listed below in black italics, while our responses and changes in manuscript are shown in blue and red, respectively.

*Pg 1, line 16: 'we analyze a multi-day photochemical smog episode.' Actually the analysis is confined to two individual days.*

Response: the original phrase has been changed as below.

"We analyze a photochemical smog episode…"

*Pg 2, line 12: remove the space between the arrow and 'RO'.*

Response: done.

*Pg 2, lines 18 – 29: Some care is needed when discussing the relative primary radical*

*sources at different urban sites: In some of the earlier studies HONO was not measured (e.g. Emmerson et al, 2005) and the inability of models to reproduce observed daytime HONO concentrations means that some of the differences observed may be caused by differences in model constraints. The model constraints used in these studies should be considered here along with the differences highlighted. Some comment on the relative source strengths in ppb/hr would be helpful and could be referred to when the results from the TC study are discussed.*

Response: we have reviewed these previous studies again. HONO was not measured in three of these earlier studies, i.e., Griffin et al. (2004) and Emmerson et al. (2005 and 2007). In the revised manuscript, we have deleted the old reference of Griffin et al. (2004; note that the measurements were conducted in 1993), and added the following statement to clarify the difference in the model constraints.

"Note that HONO was not in-situ measured but simulated by a box model in Emmerson et al. (2005 and 2007), and hence the contributions of HONO photolysis might be underestimated."

*Pg 4, line 18: Define 'MoO'.*

Response: defined.

*Pg 5, line 1: Although a reference is provided which relates to the differences between the on-line and canister alkene measurements a comment should be made in this manuscript on why the canister sample are considered more reliable than the on-line measurements.*

Response: the canister measurements had much lower detection limits (i.e., 3 pptv), while the detection limits of the real-time analyzer were much higher, especially for the alkene species with less carbon numbers. The description has been modified as follows, with changes being highlighted as the underlined sentences.

"$C_2$-$C_{10}$ non-methane hydrocarbons were measured at a time interval of 30 minutes by a commercial analyzer that combines gas chromatography (GC) with photoionization detection (PID) and flame-ionization detection (FID) (*Syntech Spectras, model GC955 Series 600/800 POCP*). The detection limits for the measured VOCs ranged from 0.001 to 0.19 ppbv. In

addition, 24-hour whole air canister samples were collected on selected days (e.g., 25 and 29 August) for the detection of $C_1$-$C_{10}$ hydrocarbons by using GC with FID, electron capture detection (ECD) and mass spectrometry detection (MSD). The analyses were carried out at the laboratory of the University of California at Irvine, and the detection limit was 3 pptv for all measured species (Simpson et al., 2010; Xue et al., 2013). As evaluated in our previous study, both sets of hydrocarbon measurements agree very well apart from the alkenes. Here the real-time data tended to systematically overestimate the canister measurements (Xue et al., 2014b). Considering the generally lower detection limit of the canister measurements, the high resolution real-time data were corrected in the present study according to the canister data."

*Section 2.2 The OBM-AOCP model: Further details on model parameters are required. The reader should not have to refer to the references for parameters specific to this study. What uptake coefficients were assumed for $HO_2$ and $N_2O_5$? What deposition rate was used? Which model species were deposited? Lines 28, 29: '...impacts on the modelling results were negligible': Which model species were considered when assessing the impact of mixing height?*

Response: in the revised manuscript, a detailed description about the model configuration and parameters has been provided in the supporting information. Below are brief responses to the specific questions.

The model adopted moderate uptake coefficients of 0.02 for $HO_2$ and of 0.014 for $N_2O_5$. The $\gamma_{N2O5}$ was taken from the observationally-derived value from our field studies in Hong Kong (Wang et al., 2016).

T. Wang, Y. J. Tham. L. K. Xue, Q. Y. Li, Q. Z. Zha, Z. Wang, C. N. Poon, W. P. Dube, D. R. Blake, P. K. K. Louie, C. W. Y. Luk, W. Tsui, S. S. Brown. Observations of nitryl chloride and modeling its source and effect on ozone in the planetary boundary layer of southern China, *J. Geophys. Res.*, 121, 5, 2016.

Dry deposition was considered for various inorganic gases and organic species such as PANs, peroxides, carbonyls and organic acids. The dry deposition velocities were adopted from the

literature of Zhang et al. (2003).

Zhang, L., Brook, J. R., and Vet, R.: A revised parameterization for gaseous dry deposition in air-quality models, *Atmos. Chem. Phys.*, 3, 2067-2082, doi: 10.5194/acp-3-2067-2003, 2003.

When assessing the impact of assumed mixing height on the modeling results, we examined the changes of $HO_X$ concentrations and OH production rates between base and sensitivity model runs. The original statement has been modified as below.

"Sensitivity model runs with different maximum mixing heights (1000 and 2000 m) indicated that its impacts on the modeling results (e.g., simulated $HO_X$ concentrations and OH production rate) were negligible."

*How was $H_2$ treated in the model? Was this measured?*

Response: $H_2$ was not measured. An initial concentration of 0.5 ppm of $H_2$ was assumed in the model. A description has been added in the revised manuscript on the $H_2$ treatment.

*Are all the VOCs listed in Table 1 used as model constraints? Does this equate to 15600 reactions or does this figure refer to the number of reactions when the MCM is run in its entirety?*

Response: yes, all the VOC species listed in Table 1 were used as model constraints. The full MCM was used in the model, and the figure "15600" refers to the number of reactions whose rates were tracked in our model. In the present study, with only a subset of primary MCM VOCs used as model constraints, the actual number of valid reactions should be smaller (the rates of some reactions should be zero). The original statement has been revised as follows.

"In our model, the rates of over 15600 reactions out of the full MCM (v3.2) are individually and instantaneously computed and grouped into a relatively small number of major routes."

*Pg 5, Line 33: It is unclear what the authors mean by '..and grouped into a relatively small number of major routes'. Does this grouping take place post-model run? I think this sentence needs re-wording so it is clear to the reader what analysis this relates to.*

Response: no, this grouping is done in the model. A module was introduced in the model to

do this. For clarity, this sentence has been modified as follows in the revised version.

"In our model, the rates of over 15600 reactions out of the full MCM (v3.2) are individually and instantaneously computed and grouped into a relatively small number of major routes."

*Pg 6, line 17: Change 'simulation' to 'comparison'*

Response: it has been rephrased to "calculation", as we don't compare modeling results with observations.

*Pg 6, line 18: what is meant by 'reasonable estimates'? Do the authors mean until steady state conditions are reached?*

Response: yes. It means the steady state conditions. This sentence has been changed as follows.

"Prior to formal calculation, the model was run for five days with constraints of the campaign-average data to reach steady states for the unconstrained compounds (*e.g.*, radicals)."

*Pg 6, line 26: The authors should consider replacing Figure 1 with the similar figure provided in the supplementary information (Figure S2) which highlights the contrasting conditions well and provides more information than presented in Figure 1.*

Response: replaced as suggested, and Figure 1 was moved to the supplementary information.

*Pg 6, line 30: Remove 'meanwhile'*

Response: done.

*Pg 6, line 33: '...intense photochemical oxidant production' this figure does not demonstrate this, rather, the subsequent model analysis reveals this.*

Response: the original sentence has been rephrased as below. The phrase "photochemical oxidant production" is changed to "photochemical pollution", as high levels of $O_3$ and PAN were observed.

"Overall, inspection of the data reveals the markedly poor air quality and serious

photochemical pollution over the region during the episode."

*Pg 7, line 3: Need to contrast these concentrations with other relevant observations, e.g. perhaps typical concentrations measured at other urban centers or could contrast with the concentrations of these species observed at TC prior to the pollution episode.*

Response: these VOC concentrations were much higher than those observed during the non-episode period. For example, the 24-h average concentrations (±SD) of toluene and xylenes during the non-episode period were 0.80±0.03 and 0.13±0.01 ppbv, compared to the levels of 9.47 and 3.87 ppbv on 25 August. The measured concentrations of formaldehyde and acetaldehyde were 3.25 and 0.83 ppbv during a non-episode day (*i.e.,* 6 September; note that we only had OVOC measurements on that day during the non-episode period), compared to 9.89 and 4.25ppbv on 25 August. The following statement has been added in the revised manuscript.

"On 25 August, for instance, the 24-h average values of toluene, summed xylenes, formaldehyde and acetaldehyde were as high as 9.47, 3.87, 9.89 and 4.25 ppbv, which were 3–30 folders higher than those measured during the non-episode period of the campaign (figures not shown)."

*Pg 7, line 6: Elevated daytime HONO concentrations at urban locations that cannot be reproduced by models which consider only gas-phase chemistry are now well reported. So not uncommon, but interesting nonetheless.*

Response: yes, we agree. The present study didn't focus on the unknown sources of daytime HONO, and only evaluated the role of HONO photolysis as a radical source.

*Pg 7, line 14: Is the Ding et al. 2004 reference appropriate given that it was published before this campaign so cannot document the source of the pollution event discussed here.*

Response: the reference has been deleted from the revised manuscript.

*Pg 7, line 20: I suggest moving figure 2 to SI. It is not critical for the paper, unlike some of the other figures currently in the SI (see later comments). Is there an order to the trajectories shown? Does Red = 25th? It would be helpful to highlight the PRD region on the map.*

Response: Figure 2 has been moved to the SI as suggested. It is further improved by labeling the trajectories and indicating the PRD region on the map. Yes, the red one is the trajectory on 25$^{th}$ August.

*Pg 7, line 22: Fig S2 does not highlight this airmass switch discussed. Should another figure be referenced here?*

Response: wind sectors have been added on this figure (note that Figure S2 has been moved into the main manuscript). To clearly show the air mass switch, back trajectories have been shown day by day throughout the measurement period in the supplementary materials of the revised paper.

*Pg 8, line 5: What is the % contribution of NO$_3$ during the day on the 31$^{st}$ August? Which VOC species is NO$_3$ primarily oxidizing? The authors suggest that ozonolysis isn't significant here because of the 'lower abundances of alkenes' so I presume carbonyls dominate NO$_3$ reactivity? More detail needed on this.*

Response: the contribution of NO$_3$ to the AOC was approximately 3% on 31$^{st}$ August. The major VOC species oxidized by NO$_3$ at TC were OVOCs and alkenes, with daytime average contributions of 77%-90% and 10%-23% respectively. The following discussion has been added in the revised manuscript.

"NO$_3$ was the second important oxidant with contributions of 7% and 3% for both cases. In particular, NO$_3$ contributed to 43% of the AOC at 15:00 LT on 25 August under a weak solar radiation condition. The major 'fuels' for NO$_3$ oxidation were OVOCs (i.e., 77%–90%) and alkenes (10%–23%)."

*How does figure 3 compare to similar analyses reported in the literature? E.g. Bannan et al. (JGR-Atmos, 2015).*

Response: the following statements have been added in the revised manuscript to compare our results with the other similar studies.

"Overall, the OH-dominated AOC at TC is in line with the previous studies at other urban locales (Elshorbany et al., 2009; Bannan et al., 2015), and the present analysis suggests that

the NO$_3$ radical may play an important role in the daytime oxidation under certain conditions (see a detailed evaluation in Section 3.4)."

*Pg 8, line 8: Is Nitryl chloride the only source of Cl atoms in the model? What was the daytime concentration of Cl atoms predicted? Please state. How was the photolysis rate of ClNO$_2$ (and the photolysis of other Cl-containing species) incorporated into the model?*

Response: yes, ClNO$_2$ photolysis was the only source of Cl atoms in the model. The modeled concentration of Cl atoms was relatively low with peak values of ~$1 \times 10^4$ atoms cm$^{-3}$, due to the moderate/low levels of ClNO$_2$ and/or weak sunlight. The photolysis frequency of ClNO$_2$ (J$_{ClNO2}$) was calculated in the model as a function of J$_{NO2}$ (J$_{ClNO2}$ = 0.04$\times$J$_{NO2}$). The treatment of photolysis of ClNO$_2$ and other Cl-containing species has been provided in the detailed model description in the revised supporting information. The statement has been modified as below in the revised manuscript.

"In comparison, O$_3$ and Cl (produced from ClNO$_2$ photolysis) had minor contributions due to the relatively lower abundances of alkenes and Cl radicals (i.e., the modeled peak value of Cl was ~$1 \times 10^4$ atoms/cm$^3$)."

*Pg 8, lines 9 and 10: poor sentence structure, please revise.*

Response: this sentence has been revised as follows.

"We further assessed the loss rates of major VOC groups due to OH oxidation, by which the partitioning of OH reactivity among different VOCs can be elucidated."

*Pg 8, line 10, figure 4: For ease of comparison with earlier reported OH reactivities at urban sites, could figure 4 be presented as a breakdown of OH reactivity with units (s-1)? i.e. divide through by modelled OH concentration.*

Response: the unit in the figure has been changed as suggested.

*Pg 8, lines 11–16: Does this partitioning include model-generated intermediate species in the OVOC segment? I note from Table 1 that elevated levels of isoprene were observed on the 25th (relative to the 31st), do the biogenic measured contribute significantly to OH reactivity at TC? How does this compare to the recent OH reactivity observations made in London*

*(Whalley et al. ACP, 2016)?*

Response: yes, the OVOC segment includes both the measured carbonyls and the modeled intermediates. Although the concentrations of isoprene on 25[th] August were higher than those on 31[st] August, they are still much lower than reactive aromatics at such an urban site. The contribution of isoprene to the OH reactivity was quite small at TC, and it was already included into the alkenes segment.

Whalley et al. (2015) reported a very interesting result that biogenic VOCs and their reaction intermediates present an important contributor of the OH reactivity in central London, and explained largely the discrepancy between measured and modeled OH reactivity. Though the roles of biogenic VOCs are different, our study agrees well with Whalley et al. (2015) on the dominance of OVOCs (including carbonyls and intermediates) in the OH reactivity of VOCs. The following statement has been put in the revised manuscript.

"These results are in fair agreement with the previous studies of Lou et al. (2010) and Whalley et al. (2015), which indicated the dominance of secondary OVOCs in the observed OH reactivity in the PRD region and central London."

*Pg 8, lines 19–34: without a direct comparison to ROx observations at TC specifically this commentary adds little to the manuscript and so I suggest removing this paragraph (or expand this section to provide further discussion of the dominant radical sources and sinks in the campaigns referred to – and how these sources and sinks compare and contrast with TC).*

Response: we agree with the point of the reviewer, and have removed this paragraph from the main manuscript (to response the other review comments, it was moved to the supplementary materials).

*Pg 9, line 3: I suggest Figs S5 and S6 are moved into the main paper.*

Response: both figures have been moved into the main paper as suggested.

*Pg 9, line 13: Some information on the identity of the other OVOCs which contribute to $HO_2$ production is needed. Are these compounds which were measured directly (e.g. acetaldehyde) or oxidation products generated by the model? How do these source strengths compare to*

*other urban locations?*

Response: these OVOC compounds include both the measured species (see Table 1) and the model-generated oxidation intermediates/products. Its source strength is comparable to those determined previously in Beijing (Liu et al., 2012) and Mexico City (Volkamer et al., 2010). The daytime-average source strengths were 2.4 ppb/hr (as a total of HCHO and other OVOCs) and 2.6 ppb/hr in Beijing and Mexico City, respectively. The following statements have been modified/added in the revised manuscript.

"For $HO_2$, the most important source is the photolysis of OVOCs (including not only the measured carbonyls but also the oxidation products generated within the model), with a daytime average production rate of 2.7 ppbv/h."

"Such source strength of OVOC photolysis was comparable to those determined in the metropolitan areas of Beijing (Liu et al., 2012) and Mexico City (Volkamer et al., 2010)."

*Pg 9, line 16: 'As to' to 'For'*

Response: changed.

*Pg 9, lines 17–18: Considering $RO_2$ deriving from $NO_3$+VOC reactions, this source seems to have a similar % contribution to the total $RO_2$ sources both on the 25th when j(NO₂) was attenuated and also on the 31st when j(NO₂) was not attenuated. A comment is needed which compares these two contrasting days.*

Response: the contribution of $NO_3$+VOC reaction pathway to the total $RO_2$ sources was 11.8% (0.2 ppb/h out of 1.7 ppb/h) on 31[st] August, which was lower than that on 25[th] August (18.5%; 0.5 ppb/h out of 2.7 ppb/h). In the revised manuscript, we have added a sub-section in Section 3.3 to discuss the radical budget on 31[st] August and compare against the local case on 25[th] August.

*Pg 9, line 20: '...different from most results obtained elsewhere' a reference to support this statement is needed.*

Response: a review paper of Stone et al. (2012), which comprehensively reviews the existing results of radical chemistry, has been cited in the revised manuscript. The original statement

has been modified as follows in the revised manuscript.

"…different from most results obtained elsewhere which have indicated the negligible role of $NO_3$ in the daytime photochemistry (Stone et al., 2012; and references therein…"

Stone, D., Whalley, L. K., and Heard, D. E.: Tropospheric OH and $HO_2$ radicals: field measurements and model comparisons, *Chem. Soc. Rev.*, 41, 6348-6404, 2012.

*Pg 9, lines 25-35: See earlier comment on model constraints in the various studies highlighted.*

Response: see the response to the earlier comment. The following statement has been added here.

"It is worth noting that HONO was not measured at Birmingham and Chelmsford but only simulated by a chemical box model, and thus the contributions of HONO photolysis were likely underestimated."

*Section 3.3: I suggest that a discussion on how the 25th and 31st contrast be included at the end of this section, focusing on local versus regional influences?*

Response: in the revised manuscript, we have added a sub-section in Section 3.3 to discuss the radical budget on 31st August and compared it with the results obtained on 25th August. The primary radical sources were essentially the same for both cases. Specifically, the major $RO_X$ sources were photolysis of OVOCs, HONO, $O_3$ and HCHO, and reactions of $O_3$+VOCs and $NO_3$+VOCs. Nevertheless, the sources of photolysis of HONO, $O_3$ and HCHO were higher on 31st August than 25th August, whilst the sources of OVOCs photolysis, $O_3$+VOCs and $NO_3$+VOCs were stronger on 25th August compared to 31st August (see the table below). This difference should be due to the higher VOC levels and attenuated solar radiation on 25th August.

Table 1. Average daytime radical sources at TC on 25th and 31st August 2011

| Source (ppb/hour) | | 25th August 2011 Hong Kong Local case | 31st August 2011 PRD Regional case |
|---|---|---|---|
| OH | HONO photolysis | 1.5 | **1.7** |

| | O$_3$ photolysis | 0.9 | **1.5** |
|---|---|---|---|
| | O$_3$+VOCs | **0.2** | 0.1 |
| HO$_2$ | Other OVOC photolysis | **1.9** | 1.3 |
| | HCHO photolysis | 0.8 | **1.2** |
| | O$_3$+VOCs | 0.1 | 0.1 |
| RO$_2$ | Other OVOC photolysis | **1.9** | 1.3 |
| | NO$_3$+VOCs | **0.5** | 0.2 |
| | O$_3$+VOCs | **0.2** | 0.1 |
| | Cl+VOCs | 0.1 | 0.1 |

*Pg 10, line 23: I am not sure that RH can be used to confirm cloud-cover. Does it matter what caused this attenuation?*

Response: the relatively low RH is indicative of little cloud on the site, but may not refer to the cloud cover at higher altitudes. The original statements have been modified as follows in the revise manuscript.

"The ambient relative humidity (RH) in the afternoon was in the range of 60%-70%, implying that there was little cloud on the site, whilst the aerosol scattering coefficient was very high (up to 525 Mm$^{-1}$; compared to 28±12 Mm$^{-1}$ on clear days). Hence, the attenuated solar radiation is possibly attributed to the abundant aerosol loadings."

*Pg 10, line 30, Fig. 9: Please include the modelled heterogeneous loss rate of N$_2$O$_5$ in this figure. What uptake of N$_2$O$_5$ was assumed?*

Response: the modelled heterogeneous loss rate of N$_2$O$_5$ has been included in the revised figure. The model adopted a moderate uptake coefficient of 0.014 for N$_2$O$_5$. The $\gamma_{N2O5}$ was taken from the average observationally-derived value from our field studies in Hong Kong (Wang et al., 2016).

T. Wang, Y. J. Tham. L. K. Xue, Q. Y. Li, Q. Z. Zha, Z. Wang, C. N. Poon, W. P. Dube, D. R. Blake, P. K. K. Louie, C. W. Y. Luk, W. Tsui, S. S. Brown. Observations of nitryl chloride and modeling its source and effect on ozone in the planetary boundary layer of southern China, *J. Geophys. Res.*, 121, 5, 2016.

*Pg 11, line 1: Does the high aerosol loading impact the heterogeneous loss of $N_2O_5$?*

Response: yes, but the model was constrained by the measured aerosol surface area data. Thus the impact of high aerosol loading on the heterogeneous loss of $N_2O_5$ had been taken into account in the present analysis.

*Section 3.5: I found this section difficult to follow as the sentence structure was poor throughout. This needs a careful edit to improve the clarity.*

Response: this section has been deleted from the revised manuscript, based on the other review comment.

*Pg 11, lines 32 – 33: '...due to the attenuated heterogeneous formation of HONO'. Were heterogeneous sources of HONO included in the model? If they were, a description of these sources should be included in the experimental description. If they were not included, then attenuated heterogeneous formation of HONO cannot be the cause of the reduction of OH in the model. Is this trend not simply caused by a reduction in the secondary OH source from $HO_2+NO$? In this analysis are NO and $NO_2$ concentrations both reduced by the same fraction?*

Response: we are sorry that we didn't clearly state this, which made the manuscript confusing. Our model includes the heterogeneous formation of HONO from reactions of $NO_2$ on ground and aerosol surfaces. In the revised manuscript, a detailed description of the model has been provided in the supplementary materials. As stated above, anyway, this section has been deleted from the revised version.

*Why were only alkanes, alkenes and aromatics considered in this analysis? What is the sensitivity to the other VOC groups measured, e.g. the Biogenics?*

Response: as stated above, the sensitivity studies have been deleted in the revised manuscript.

*Conclusions: This section should provide some commentary on the local versus regional events; the latter sections of the paper neglect the case study on the 31st and it would conclude the paper nicely if the results from this day were evaluated alongside the 25th here.*

Response: in the revised manuscript, we have added a discussion of the case study on 31st

August (see the responses above). The following sentences have also been added in the conclusion section to state the difference between the two cases.

"Higher AOC levels and stronger primary production of radicals were determined during the Hong Kong local case compared to the PRD regional case. Although the primary radical sources were essentially the same, photolysis of OVOCs (except for HCHO) and reactions of $O_3$+VOCs and $NO_3$+VOCs were stronger for the local case, which was ascribed to the higher VOC levels. In comparison, the source strengths of photolysis of HONO, $O_3$ and HCHO were higher during the regional case."

---

## Referee Report (RR1)

Pg 1, line 27 change to 'in one case when solar irradiation was attenuated, possibly by the..'

Pg 6, line 26 remove 'obviously'

Pg 8, line 19 'by which' to 'from which'

Fig 6a 'HONO Photolysis'